# THE ALIGNMENT BOTTLENECK

## ABSTRACT

We study feedback alignment for large language models under a finite information budget. The feedback loop is modeled as a two-stage channel $U \to H \to Y$ given context $S$, where $U$ is the target, $H$ is the bounded judgment, and $Y$ is the label. The average capacity $\bar{C}_{\text{tot}|S}$ of this channel constitutes an alignment bottleneck. By applying Fano's inequality to separable codebooks, we derive a minimax lower bound on alignment error that depends on value complexity $\log M$ and capacity but is independent of dataset size. This implies that scaling data cannot eliminate error when the feedback channel is structurally deficient. We further show that the same capacity term controls the environmental budget in a PAC-Bayes generalization bound. These results define a performance interval where optimization beyond the channel capacity fits rater artifacts such as sycophancy. Experiments with Qwen confirm that low-capacity feedback leads to saturation and degradation even as data scales. Our framework suggests that improving alignment requires increasing the channel capacity through richer interfaces or clearer constitutions rather than just collecting more data.

## 1 INTRODUCTION

Scaling laws keep improving LLM capabilities (Kaplan et al., 2020; Wei et al., 2022; Hestness et al., 2017), but feedback-based alignment shows a tension: instruction following improves on average, while systematic deviations from intended behavior persist. In practice, pipelines specify desirable behavior through constitutions, safety policies, task rubrics, reward guidelines or looser rater instructions, and then train on human or AI feedback given as preferences, scores or critiques (Ouyang et al., 2022; Bai et al., 2022b; Ziegler et al., 2020; Zheng et al., 2023; Rafailov et al., 2024b; Lee et al., 2024). The specification can be rich and high dimensional, while each label is cheap, so the data pipeline transmits only a small and noisy slice of the underlying target $U$, whether $U$ is written as explicit rules, encoded in a reward model, or held implicitly by raters.

Despite large gains, aligned models still show sycophancy, reward hacking, inverse scaling on truthfulness, and sequential risks such as user tampering (Perez et al., 2023; Sharma et al., 2024; Pan et al., 2022; Denison et al., 2024; Lin et al., 2022; Evans and Kasirzadeh, 2023). These patterns arise under both human and AI feedback and persist when models and datasets are scaled. This suggests that some failures may reflect a structural limit of the feedback loop rather than only algorithmic or architectural details.

Our starting point is that feedback is produced by bounded agents. Human raters and model raters that emulate them operate under limited time, attention and working memory. Work on bounded rationality and rate distortion views decisions as resource limited and satisficing, with people compressing task representations and trading performance for cognitive cost (Simon, 1955; Lewis et al., 2014; Ho et al., 2022; 2020; Zénon et al., 2019; Zaslavsky et al., 2021). When raters cannot fully apply the target specification on each example, labels cannot be treated as ground truth plus small noise; they are the output of a finite capacity channel from latent targets $U$ to logged feedback $Y$.

We model the feedback loop as a two-stage communication cascade $U \to H \to Y$ given context $S$. The latent variable $U$ is a finite codebook of task targets (for a given context-response pair it can encode which normative rules hold, or index rewards, rankings, or coarse labels), $H$ is the rater's bounded judgment under time and cognitive constraints, and $Y$ is the observable label (such as a preference, score, or critique) given $S$. In rule-based specifications, $U$ can record a pattern of rule satisfaction; a high-capacity rater attempts to apply many rules before producing $Y$, while

a bottlenecked rater compresses them into cheap heuristics such as tone or agreement, effectively projecting $U$ to a lower-dimensional signal. We use rule-based illustrations only to make value complexity concrete; the analysis applies to any feedback alignment scheme that communicates a latent target $U$ through labels $Y$, regardless of how $U$ is represented.

We then establish a capacity link: the same quantity that limits value information entering the data also controls the statistical complexity needed for generalization. On the lower bound side (Sec. 4), using separable codebooks and Fano's inequality, we obtain an information lower bound on true risk that is independent of dataset size

$$R_{\mathrm{mix}}(\pi) \;\geq\; (\varepsilon + \Delta)\left(1 - \frac{\bar{C}_{\mathrm{tot}|S}^{\mathrm{mix}} + \log 2}{\log M}\right)_+.$$

On the upper bound side (Sec. 5), using PAC–Bayes for bounded observable losses and the link between KL complexity and dataset parameter mutual information (Xu and Raginsky, 2017; Russo and Zou, 2019; Rodríguez-Gálvez et al., 2024; Lotfi et al., 2024; Dziugaite and Roy, 2017), we show

$$\mathbb{E}_{\mathcal{D}}\big[\mathrm{KL}(P\|Q)\big] \;\leq\; m\,\bar{C}_{\mathrm{tot}|S} \;+\; m\,I(U;S) \;+\; \rho \;+\; \mathrm{KL}\big(p(\theta)\|Q\big),$$

which turns the PAC–Bayes KL term into an explicit environmental budget set by the same feedback channel that limits $I(U;Y \mid S)$. Under the canonical observable loss and the same codebook mixture, this yields a capacity-coupled interval for the true risk of the randomized policy $\pi_\theta$:

$$(\varepsilon + \Delta)\left(1 - \frac{\bar{C}_{\mathrm{tot}|S}^{\mathrm{mix}} + \log 2}{\log M}\right)_+ \;\leq\; \mathbb{E}_{\theta \sim P}\big[R_{\mathrm{mix}}(\pi_\theta)\big] \;\leq\; \mathbb{E}_{\theta \sim P}\big[\widehat{R}_m^{\mathrm{obs}}(\theta)\big] + \sqrt{\frac{\mathrm{KL}(P\|Q) + \log(1/\delta)}{2m}},$$

with the expected KL term itself controlled by $\bar{C}_{\mathrm{tot}|S}$. Conceptually, we treat the feedback loop as a bounded channel $U \to H \to Y$ and use its average capacity $\bar{C}_{\mathrm{tot}|S}$ as an explicit alignment bottleneck. Technically, the same bottleneck induces a data-size-independent lower bound on alignment error and a PAC–Bayes generalization upper bound through an environmental budget of order $m\,\bar{C}_{\mathrm{tot}|S} + m\,I(U;S) + \rho$, so $\bar{C}_{\mathrm{tot}|S}$ largely determines both the error floor and the optimization ceiling. Finally, a DPO experiment shows capacity saturation and overfitting to rater-specific quirks. The lower bound holds for any decoder from $(Y, S)$ to actions and is independent of model parameterization.

These bounds imply that increasing the dataset size $m$ cannot beat the lower bound when separability and capacity are fixed, that reaching a target risk requires capacity that scales with value complexity (which constrains pluralistic or multi-objective alignment), and that once useful signal saturates capacity, further optimization mainly fits residual channel regularities, in line with observed sycophancy and reward hacking (Perez et al., 2023; Sharma et al., 2024). Throughout we use alignment bottleneck to refer to the average total capacity $\bar{C}_{\mathrm{tot}|S}$ of the feedback process, induced by bounded rationality and interface design, which both limits how far alignment can go and shapes where additional optimization pressure flows, regardless of whether the specification is written as a constitution, implemented as a reward model, or held implicitly by raters.

## 2 RELATED WORK

**Feedback Alignment and Systematic Deviations.** Feedback-based alignment trains policies on preference-like signals collected under various supervision protocols (Christiano et al., 2017; Ouyang et al., 2022; Ziegler et al., 2020; Bai et al., 2022a;b; Zheng et al., 2023; Rafailov et al., 2024b; Ethayarajh et al., 2024; Lin et al., 2024b; Lee et al., 2024; Guo et al., 2024; Mu et al., 2024). Despite strong gains, models still show sycophancy, reward hacking, inverse scaling on truthfulness, and sequential risks such as user tampering (Perez et al., 2023; Sharma et al., 2024; Pan et al., 2022; Denison et al., 2024; Lin et al., 2022; Evans and Kasirzadeh, 2023). System-level analyses explain these effects via incentives, oversight gaps and aggregation protocols (Irving et al., 2018; Everitt et al., 2021; Ge et al., 2024; Ngo et al., 2024; Rane et al., 2024), and recent work studies optimization induced misalignment and scaling of proxy rewards and KL budgets (Rafailov et al., 2024a; Mroueh and Nitsure, 2025). Our work is complementary: the theoretical analysis fixes a given supervision protocol and studies a static source channel model of the training feedback, where a single capacity term $\bar{C}_{\mathrm{tot}|S}$ controls both a Fano lower bound and the PAC–Bayes complexity.

**Bounded Rationality, Alignment, and Cognitive Constraints.** Classical bounded rationality views decisions as resource limited and satisficing (Simon, 1955). Cognitive and information theoretic

models make this precise using constrained internal state, rate distortion and policy compression (Lewis et al., 2014; Ortega and Braun, 2011; Zénon et al., 2019; Sims, 2016; Zaslavsky et al., 2021; Ho et al., 2022; 2020; Gottwald and Braun, 2019; Arumugam et al., 2023; Arumugam and Van Roy, 2021; Arumugam and Roy, 2021; 2022; Lai and Gershman, 2021). Recent work brings these ideas to LLM alignment, for example satisficing alignment at inference time (Chehade et al., 2025), learning under unreliable or resource limited supervision (Hyland et al., 2024), explicit reasoning budgets (Lieder and Griffiths, 2020), and critiques of preference based alignment under cognitive limits (Zhi-Xuan et al., 2024). These papers mainly design new decision rules or supervision schemes given bounded agents. Here we instead treat bounded rationality as the mechanism that induces a finite $I(U; H \mid S)$ in the training loop, which through the cascade $U \to H \to Y$ bounds $I(U; Y \mid S)$ and defines the average total capacity $\bar{C}_{\text{tot}|S}$ of the feedback channel. The analysis keeps the decoder family fixed and surfaces the information budget that existing feedback alignment runs implicitly operate under.

**Information Theoretic Generalization and PAC–Bayes.** Information theory has been used to study representation and generalization via the Information Bottleneck, rate distortion style converses and related notions of compression (Tishby et al., 2000; Tishby and Zaslavsky, 2015; Alemi et al., 2017; Kolchinsky et al., 2019; Peng et al., 2020; Shannon, 1959; Ngampruetikorn and Schwab, 2022; Saxe et al., 2019; Goldfeld et al., 2019; Shwartz-Ziv and LeCun, 2023; Shwartz-Ziv and Tishby, 2017; Kawaguchi et al., 2023; He et al., 2025; Bartlett et al., 2017; Shwartz-Ziv et al., 2024). PAC–Bayes theory gives non asymptotic generalization bounds that remain meaningful in large models and has been linked to dataset parameter mutual information (Langford and Caruana, 2001; Neyshabur et al., 2017; Dziugaite and Roy, 2017; Neyshabur et al., 2018; Lotfi et al., 2024; Rodríguez-Gálvez et al., 2024; Wu et al., 2025; Leblanc et al., 2025; Picard-Weibel et al., 2025; Xu and Raginsky, 2017; Russo and Zou, 2019; Wang et al., 2022). Our contribution is to analyze both sides of the problem with respect to the same concrete feedback channel $U \to H \to Y$: on the converse side we build a data-size-independent wall on true risk in terms of value complexity $\log M$ and $\bar{C}_{\text{tot}|S}$, and on the generalization side we show that the PAC–Bayes KL term is upper bounded by an environmental budget $m\,\bar{C}_{\text{tot}|S} + m\,I(U; S) + \rho + \text{KL}(p(\theta)\|Q)$. To our knowledge, prior work has not explicitly coupled a Fano style lower bound and a PAC–Bayes upper bound through a single channel capacity term interpreted as a human or rater feedback bottleneck.

## 3 PROBLEM SETUP

We formalize the Alignment Bottleneck as a resource-constrained inference and communication problem. A specification of desired behavior enters as a latent target $U$ (a finite codebook of configurations such as rule satisfaction patterns, reward bins, or other task targets), and the human or AI rater is modeled as a two-stage channel with a finite information budget that will control both the lower and upper bounds.

### 3.1 TASK, LOSS, AND FEEDBACK CHANNEL

We first define a generic alignment task, with $S$ as context (prompt and metadata), $U$ as the latent target, $Y$ as feedback, and $\pi$ as the learned decoder.

**Definition 1** (Task and Observable Loss)**.** *Let $S$ denote publicly observable context, $U$ the latent task target ("what humans truly want"), and $Y$ the human feedback emitted through a finite-capacity channel. A learner outputs an action $\hat{a} = \pi(Y, S) \in \mathcal{A}$ using a decoder $\pi$. The task loss is a bounded measurable function $\ell : \mathcal{U} \times \mathcal{A} \to [0, 1]$ (such as 0–1 loss, pairwise ranking loss mapped to $[0, 1]$, or a truncated and normalized MSE; see Appendix H). The (population) risk is*

$$R(\pi) \triangleq \mathbb{E}\big[\ell(U, \pi(Y, S))\big]. \tag{1}$$

In a constitutional example, $\ell(U, \pi(Y, S))$ is a normalized count of violated rules on a context–response pair, but $U$ can equally encode rewards, rankings, or coarse labels, and $\ell$ can be any bounded misalignment loss. The truncation to $[0, 1]$ is only for keeping later generalization bounds clean and does not assume that values are explicitly written down.

**Definition 2** (Human Channel Families)**.** *We model the human-in-the-loop communication by a cascade $U \to H \to Y$ given $S$. The cognitive stage uses a conditional kernel $p(h \mid u, s) \in \mathcal{F}_{\text{cog}}$, and the articulation stage uses $p(y \mid h, s) \in \mathcal{F}_{\text{art}}$. The learner observes only $(Y, S)$, not $H$.*

Here $\mathcal{F}_{\text{cog}}$ collects how raters, under bounded cognition, map $(U, S)$ into an internal judgment $H$, and $\mathcal{F}_{\text{art}}$ collects how $H$ is turned into labels $Y$ (binary preferences, scores, critiques, etc.) through the interface. Constitutions or rule lists are one convenient way to define $U$ in practice because they make value complexity explicit, but the same formalism applies when $U$ is only implicit in instructions or reward models.

Empirical and theoretical work in cognitive science and decision making shows that human judgment is resource-bounded and compressive (Lewis et al., 2014; Zénon et al., 2019; Ortega and Braun, 2011; Gottwald and Braun, 2019; Ho et al., 2022; Sims, 2016; Zaslavsky et al., 2021; Lai and Gershman, 2021; Arumugam and Van Roy, 2021; Arumugam and Roy, 2021; 2022; Arumugam et al., 2023). We use this as the mechanism behind a finite $I(U; H \mid S)$ and treat the cognitive capacity $C_{\text{cog}|S}$ as a structural bottleneck through which value information must pass, regardless of whether the specification is rule-based or implicit.

**Assumption 1** (Per-stage Feasibility). *All admissible systems considered in this paper satisfy $p(h \mid u, s) \in \mathcal{F}_{\text{cog}}$ and $p(y \mid h, s) \in \mathcal{F}_{\text{art}}$ for almost every $(u, s)$, with the two stages independent across i.i.d. samples.*

This captures the usual feedback-based alignment picture: each label is produced independently by first forming $H$ under bounded cognition, then articulating $Y$ through the interface.

## 3.2 Context-Conditional Capacities

We define the per-stage capacities and the total capacity of the cascade.

**Definition 3** (Cognitive Capacity). *For each s, define*

$$C_{\text{cog}|S}(s) \triangleq \sup_{p(h|u,s) \in \mathcal{F}_{\text{cog}}} I(U; H \mid S = s). \tag{2}$$

**Definition 4** (Articulation Capacity). *For each s, define*

$$C_{\text{art}|S}(s) \triangleq \sup_{p(y|h,s) \in \mathcal{F}_{\text{art}}} I(H; Y \mid S = s). \tag{3}$$

**Definition 5** (Total Capacity and Its Average). *For each s, define the per-context total capacity*

$$C_{\text{tot}|S}(s) \triangleq \min \left\{ C_{\text{cog}|S}(s), \, C_{\text{art}|S}(s) \right\}, \tag{4}$$

*and its average*

$$\bar{C}_{\text{tot}|S} \triangleq \mathbb{E}_S \left[ C_{\text{tot}|S}(S) \right]. \tag{5}$$

Thus $C_{\text{cog}|S}(s)$ is the maximum information about $U$ that can enter $H$ in context $s$, $C_{\text{art}|S}(s)$ is the maximum information about $H$ that can be expressed as $Y$, and $C_{\text{tot}|S}(s)$ is their minimum. The average $\bar{C}_{\text{tot}|S}$ is the central quantity of the paper and will be our formal notion of the alignment bottleneck: an information budget on how much of the target $U$ can reach the learner through feedback.

**Proposition 1** (Cascade Upper Bound via Data Processing). *Under Assumption 1, any admissible cascade $U \to H \to Y$ forms a Markov chain. By the data processing inequality (Shannon, 1948), it satisfies for every s,*

$$I(U; Y \mid S = s) \leq \min\{I(U; H \mid S = s), \, I(H; Y \mid S = s)\} \leq C_{\text{tot}|S}(s), \tag{6}$$

*and hence, averaging over S,*

$$I(U; Y \mid S) \leq \bar{C}_{\text{tot}|S}. \tag{7}$$

Technical details about source-dependent capacities and coarsenings of $S$ are deferred to Appendix L.

## 4 Information-Theoretic Lower Bounds

We derive the first component of the Alignment Performance Interval: an information-theoretic lower bound on the true risk that couples value complexity and channel capacity. Following the classical minimax recipe (Shannon, 1948), we build a family of hard but distinguishable tasks and apply Fano's inequality, obtaining a converse that holds for any decoder $\pi$ mapping $(Y, S)$ to actions, independent of parameterization or model class.

### 4.1 Separable Codebooks and the Loss–Index Link

We summarize the key construction and defer formal details to Appendix D.1. A $\Delta$-separable codebook is a finite collection $\{(u^{(i)}, a^{(i)})\}_{i=1}^{M}$ such that each $a^{(i)}$ is good for its own target $u^{(i)}$ (loss at most $\varepsilon$) and incurs at least an extra margin $\Delta$ when used for any other index. A measurable decoder $\phi$ maps actions to indices so that decoding to the wrong index implies loss at least $\varepsilon + \Delta$ (Assumption 2). This reduces alignment error to an index-identification problem suitable for Fano.

Operationally, one can construct such a codebook from any way of specifying good behavior: let $U$ index finitely many distinguishable rule satisfaction patterns, reward bins, or other targets, choose $a^{(i)}$ tuned to each pattern, and use a simple decoder $\phi$ (for example nearest prototype or predicted class). The resulting $M$ and $\Delta$ summarize value complexity. In practice, these templates give a checklist: instantiate $\{(u^{(i)}, a^{(i)})\}$ from the task specification, pick $\phi$, and verify on a strong evaluator that mis-decoded actions typically incur loss at least $\varepsilon + \Delta$ (or the soft variant in Appendix M).

### 4.2 Fano–Packing Converse Lower Bound

**Theorem 1** (Fano–Packing Lower Bound (Bayes/Minimax Semantics)). *Let $\ell \in [0, 1]$ and suppose there exists a $\Delta$-separable codebook of size $M$ satisfying Assumption 2. Let $J \sim \mathrm{Unif}[M]$, and define the mixture distribution over $(U, S)$ by setting $U = U^{(J)}$, with $U^{(J)}$ measurable in $(J, S)$, and $S \sim P(S)$. Assume $M \geq 2$. Write $R_{\mathrm{mix}}(\pi)$ for the risk under this mixture distribution. Then, for any decoder $\pi$,*

$$R_{\mathrm{mix}}(\pi) \; \geq \; (\varepsilon + \Delta) \left( 1 - \frac{I(U; Y \mid S) + \log 2}{\log M} \right)_{+}. \tag{8}$$

*In particular, using equation 7,*

$$R_{\mathrm{mix}}(\pi) \; \geq \; (\varepsilon + \Delta) \left( 1 - \frac{\bar{C}_{\mathrm{tot}|S}^{\mathrm{mix}} + \log 2}{\log M} \right)_{+}. \tag{9}$$

*Equivalently, these yield a standard minimax lower bound over the family of sources supported on the codebook.*

The proof follows a standard Fano argument via two lemmas, deferred to Appendix E.

### 4.3 Capacity-Limited Achievability and the Information Wall

The lower bound can be summarized as an information wall that depends on both value complexity and feedback capacity: for a problem class $\mathcal{P}$ of admissible codebooks, as $\log M$ grows while $\bar{C}_{\mathrm{tot}|S}^{\mathrm{mix}}$ is fixed, the wall rises and no learner can push $R_{\mathrm{mix}}(\pi)$ below it. The wall is also empirically testable probeable: once a specification and feedback interface are fixed, one can use a strong evaluator that applies the full spec to approximate a codebook mixture, estimate $M$ and $\Delta$ from how many target configurations the evaluator can reliably distinguish, and then observe how performance saturates under the production feedback channel, as in the synthetic experiment of Sec. 7.3. The formal definition and minimax interpretation are given in Appendix E.1.

## 5 PAC–Bayes Upper Bounds

We use PAC–Bayes, a non-asymptotic framework suited to overparameterized learners (including LLMs) with non-vacuous guarantees at scale (Lotfi et al., 2024). Our aim is to make its complexity term, $\mathrm{KL}(P\|Q)$, capacity-explicit in the same $\bar{C}_{\mathrm{tot}|S}$ that drives the converse, closing the loop between the lower wall and the statistical ceiling. On this upper-bound side we instantiate the decoder as a randomized policy $\pi_\theta$ drawn from a posterior; this parameterization is convenient for PAC–Bayes, but the lower-bound analysis in Sec. 4 did not rely on any such choice.

### 5.1 PAC–Bayes Bounds

We recall a standard PAC–Bayes result for bounded losses as the basis of the ceiling. Recent variants tighten constants, cover heavier tails, and allow anytime validity, yielding non-vacuous bounds even

for billion-parameter LLMs (Dziugaite and Roy, 2017; Rodríguez-Gálvez et al., 2024; Lotfi et al., 2024; Wu et al., 2025). Tightness is not automatic: strong guarantees require priors that put sufficient mass on high-performing predictors (Picard-Weibel et al., 2025). This interacts with the Alignment Bottleneck: finite human-feedback capacity limits how informative data-independent priors can be, and this constraint enters through the KL term that we bound via $\bar{C}_{\text{tot}|S}$.

**Theorem 2** (PAC–Bayes for Observable Loss). *Let $\tilde{\ell} : \mathcal{Y} \times \mathcal{S} \times \mathcal{A} \to [0,1]$ be any bounded loss measurable with respect to the observed data $(Y, S)$. For any $\delta \in (0,1)$, with probability at least $1 - \delta$ over the i.i.d. draw of the dataset $\mathcal{D} = \{(Y_i, S_i)\}_{i=1}^m$,*

$$\mathbb{E}_{\theta \sim P}\big[R_{\text{obs}}(\theta)\big] \ \leq \ \mathbb{E}_{\theta \sim P}\big[\widehat{R}_m^{\text{obs}}(\theta)\big] \ + \ \sqrt{\frac{\text{KL}(P\|Q) + \log(1/\delta)}{2m}}, \tag{10}$$

*where*

$$R_{\text{obs}}(\theta) \ \triangleq \ \mathbb{E}\big[\tilde{\ell}\big(Y, S, \pi_\theta(Y, S)\big)\big], \qquad \widehat{R}_m^{\text{obs}}(\theta) \ \triangleq \ \frac{1}{m} \sum_{i=1}^m \tilde{\ell}\big(Y_i, S_i, \pi_\theta(Y_i, S_i)\big).$$

*Canonical choice.* If we choose $\tilde{\ell} = \tilde{\ell}^\star$ as in Appendix J, then $R_{\text{obs}}(\theta) = R(\pi_\theta)$ holds for the same data distribution.

## 5.2 KL Decomposition and Capacity Control

We summarize the capacity-explicit control of the PAC–Bayes KL term and defer technical lemmas and proofs to Appendix F.1. Combining an expected KL decomposition with a dataset–parameter information bound and a residual-information assumption yields the following capacity-explicit bound for the posterior complexity.

**Theorem 3** (Capacity-explicit PAC–Bayes complexity). *Under the assumptions of Section 3, and with a prior $Q$ independent of the dataset, the expected PAC–Bayes complexity satisfies*

$$\mathbb{E}_{\mathcal{D}}\big[\text{KL}(P\|Q)\big] \ \leq \ m\,\bar{C}_{\text{tot}|S} \ + \ m\,I(U;S) \ + \ \rho \ + \ \text{KL}\big(p(\theta)\,\|\,Q\big),$$

*where $\bar{C}_{\text{tot}|S}$ is the average total capacity of the feedback channel, $I(U;S)$ quantifies source–context dependence, and $\rho$ is a residual term capturing information about the particular sample beyond $U^m$. The precise statement and derivation are given in Corollary 2 in Appendix F.1.*

This turns the PAC–Bayes KL term into an explicit environmental budget set by the same feedback channel that limits $I(U; Y \mid S)$. The KL complexity cannot grow arbitrarily with more data: the useful part $I(U^m; \theta)$ is capped by $m\,\bar{C}_{\text{tot}|S} + m\,I(U;S)$, and any additional fitting must go into the residual $I(\mathcal{D}; \theta \mid U^m)$, captured by $\rho$. In particular, once the channel has saturated, further decreases in empirical observable loss $\widehat{R}_m^{\text{obs}}$ must correspond to the learner encoding more of the residual quirks of the feedback process rather than additional information about $U$.

## 5.3 Algorithmic Residual Information

We interpret the residual term $\rho$ in Theorem 3 as measuring information that the learning algorithm encodes about the particular dataset beyond the latent values $U^m$, such as overfitting to channel noise or rater quirks. Assumption 3 in Appendix F.1 formalizes this notion and shows how $\rho$ can be controlled via algorithmic noise, early stopping, or posterior smoothing. Practically, a data-independent randomized compression of the posterior can enforce a finite residual without increasing $\text{KL}(P\|Q)$ (Appendix K), giving a concrete way to keep the PAC–Bayes complexity near the capacity-implied environmental budget.

## 6 The Alignment Performance Interval

We developed two components: an information-theoretic error floor via Fano's inequality (Section 4) and a statistical error ceiling via PAC–Bayes theory (Section 5). We establish the Alignment Performance Interval. The same capacity term (the channel capacity $\bar{C}_{\text{tot}|S}$) determines the lower bound and, at the same time, limits the learnable model complexity that determines the generalization upper bound.

## 6.1 Capacity-Coupled Bounds

Let $\mathcal{P}$ be a collection of codebooks $\mathcal{C}(M, \Delta, \varepsilon)$. For any learning algorithm (decoder) $\pi$, its worst-case true risk under a codebook-induced mixture distribution is bounded from below by the information-theoretic wall:

$$\textbf{Lower (Minimax):} \qquad \sup_{\mathcal{C} \in \mathcal{P}} R^{\mathcal{C}}_{\mathrm{mix}}(\pi) \; \geq \; \mathsf{Wall}\big(\bar{C}^{\mathrm{mix}}_{\mathrm{tot}|S}; \mathcal{P}\big) \quad \text{from Thm. 1.} \qquad (11)$$

Simultaneously, for any prior $Q$ and posterior $P$, the expected true risk is bounded from above. With probability $\geq 1 - \delta$ over the draw of a dataset $\mathcal{D}$ from the same mixture, and using the canonical observable loss $\tilde{\ell}^{\star}$ (Appendix J) such that $R_{\mathrm{obs}}(\theta) = R_{\mathrm{mix}}(\pi_\theta)$, we have:

$$\textbf{Upper (High-probability):} \qquad \mathbb{E}_{\theta \sim P}\big[R_{\mathrm{mix}}(\pi_\theta)\big] \; \leq \; \mathbb{E}_{\theta \sim P}\big[\widehat{R}^{\mathrm{obs}}_m(\theta)\big] + \sqrt{\frac{\mathrm{KL}(P\|Q) + \log(1/\delta)}{2m}}.$$
$$(12)$$

This follows from Theorem 2 applied with the canonical observable loss $\tilde{\ell}^{\star}$ that matches the true risk $R_{\mathrm{mix}}$ (Appendix J). As shown in Corollary 2, the expected KL-divergence term is controlled by the channel capacity, $\mathbb{E}_{\mathcal{D}}\big[\mathrm{KL}(P\|Q)\big] \; \leq \; m\,\bar{C}_{\mathrm{tot}|S} \; + \; m\,I(U; S) \; + \; \rho \; + \; \mathrm{KL}\big(p(\theta)\,\|\,Q\big)$, thus explicitly coupling the ceiling to the same capacity term that defines the floor. Together, equation 11 and equation 12 yield two-sided bounds on the same risk quantity, $R_{\mathrm{mix}}$, driven by $\bar{C}_{\mathrm{tot}|S}$.

Interpretation. The two bounds control different risks: the Bayes/minimax lower bound applies to the true risk under the mixture distribution $R_{\mathrm{mix}}$, whereas the PAC–Bayes upper bound applies to the observable risk $R_{\mathrm{obs}}$ under the actual data distribution. Without an explicit link between $\ell$ and $\tilde{\ell}$ and without a distribution match, they should not be treated as an interval on the same quantity. Under a mild link between the observable loss and the true alignment loss (formalized in Appendix I) and when $\mathcal{D}$ is drawn from the same codebook-induced mixture used in Thm. 1, the PAC–Bayes ceiling can be transferred directly to $R_{\mathrm{mix}}$ up to task-dependent constants. With a canonical observable loss and matched data distribution (Appendix J), both the lower and upper bounds apply to the same risk $R_{\mathrm{mix}}$, yielding a genuine two-sided performance interval.

In alignment terms, for any feedback alignment scheme that can be summarized by a codebook and a feedback channel, the same scalar capacity $\bar{C}_{\mathrm{tot}|S}$ jointly controls a floor and a ceiling on the achievable true alignment error. For a fixed specification $U$ and feedback channel, this means $R_{\mathrm{mix}}$ is squeezed between a floor determined by how much of the specification can traverse the channel and a ceiling determined by how much of that information the learner encodes in its parameters.

## 7 Implications for Alignment Design

The Alignment Performance Interval (Sec. 6) explains practical alignment limits and suggests design levers. This section focuses on what the three main implications mean for practitioners running alignment pipelines, interpreting the mathematical statements in terms of concrete choices about specifications, feedback, and optimization.

### 7.1 Implication I: Data Size Independent Lower Bound

**Corollary 1** (Information-theoretic lower bound independent of $m$). *Let $\mathcal{C}(M, \Delta, \varepsilon)$ be any $\Delta$-separable codebook with $M \geq 2$, and let $R_{\mathrm{mix}}$ denote the risk under its mixture distribution (as in Thm. 1). For any decoder $\pi$,*

$$R_{\mathrm{mix}}(\pi) \; \geq \; (\varepsilon + \Delta)\left(1 - \frac{\bar{C}^{\mathrm{mix}}_{\mathrm{tot}|S} + \log 2}{\log M}\right)_{+}, \qquad (13)$$

*which is exactly equation 9. The bound equation 13 does not depend on $m$, hence the lower bound is independent of dataset size.*

From an alignment perspective, this says that once the specification codebook and the feedback channel are fixed, there is a risk floor that does not move with more data: if the average total capacity $\bar{C}^{\mathrm{mix}}_{\mathrm{tot}|S}$ is small relative to the value complexity $\log M$, then even taking $m \to \infty$ cannot push $R_{\mathrm{mix}}(\pi)$ below the wall. In a concrete alignment pipeline, holding fixed the value spec (whether

a short policy, a long constitution, or implicit guidelines), rater pool, and interface while scaling the number of preference pairs or reward model examples will eventually enter a regime where specification-level metrics flatten and only fluctuate around a nonzero error level.(Lin et al., 2024a; Korkmaz et al., 2025; Lin et al., 2022). When teams see such saturation, especially under a strong evaluator that actually checks the full spec, it is more natural under this view to diagnose a capacity wall of the feedback system than to blame optimizer details or model size; changing the wall requires changing the channel or the specification, not just collecting more of the same labels. For current feedback-alignment-style setups with short prompts and a few seconds of rater time per label, it is plausible that the effective capacity per example is quite small—on the order of a handful of bits—consistent with bounded working-memory estimates; Appendix B discusses how to empirically calibrate such capacities rather than assume them.

## 7.2 IMPLICATION II: CAPACITY REQUIREMENTS FOR TARGET RISK

**Proposition 2** (Necessary capacity for a target risk). *Fix a codebook $\mathcal{C}(M, \Delta, \varepsilon)$ and a target risk $r \in [0, 1]$. If a decoder $\pi$ satisfies $R_{\mathrm{mix}}(\pi) \leq r$, then necessarily*

$$\bar{C}^{\mathrm{mix}}_{\mathrm{tot}|S} \geq \left(1 - \frac{r}{\varepsilon + \Delta}\right) \log M - \log 2. \tag{14}$$

*Proof.* Rearrange equation 9; the $(\cdot)_+$ can be dropped once $r < \varepsilon + \Delta$ (otherwise the inequality is vacuous but true). $\square$

Here $\log M$ plays the role of value complexity: for a fixed way of specifying what counts as good behavior, it measures how many distinct "target patterns" the system is supposed to distinguish. The inequality shows that keeping a target risk $r$ fixed while increasing $M$ forces a proportional increase in $\bar{C}^{\mathrm{mix}}_{\mathrm{tot}|S}$. In practice, if the implicit or explicit value specification grows more complex (more rules, more edge cases, more objectives), while raters still give only binary preferences or short scores under the same time budget, then the required capacity is not available and the achievable alignment floor moves up.(Sorensen et al., 2024; Guo et al., 2024; Fisher et al., 2025) For an alignment engineer, this points to two concrete levers when a specification becomes richer: either decompose it into smaller modules so that each dataset has a smaller effective codebook, or deliberately increase feedback capacity per example by switching from single-bit preferences to multi-aspect labels, structured rubrics, richer rating scales, or longer critiques (possibly via stronger AI raters), so that each observed $Y$ carries more information about the configuration $U$.(Shannon, 1959) This applies whether the value specification is written as a constitution, implemented as a hidden reward model, or only reflected in ad-hoc rater instructions.

## 7.3 IMPLICATION III: CAPACITY CONTROLLED COMPLEXITY AND CHANNEL OVERFITTING

**Theorem 4** (Capacity-controlled PAC–Bayes complexity). *Under the i.i.d. source and memoryless channel, with a prior $Q$ independent of $\mathcal{D}$ and any learning algorithm whose residual satisfies Assumption 3, the expected PAC–Bayes complexity obeys*

$$\mathbb{E}_{\mathcal{D}}\big[\mathrm{KL}(P\|Q)\big] \leq m\,\bar{C}_{\mathrm{tot}|S} + m\,I(U; S) + \rho + \mathrm{KL}\big(p(\theta)\|Q\big), \tag{15}$$

*as given in Cor. 2. Combining equation 15 with the Markov lift in Appendix G and equation 12 yields a high-probability* capacity-explicit *upper bound on the (observable or, under the canonical choice, true) risk.*

*An Information-Theoretic View of Overfitting to the Channel.* When $\widehat{R}^{\mathrm{obs}}_m(\theta)$ is driven close to zero but a small $\bar{C}_{\mathrm{tot}|S}$ keeps the Fano floor nontrivial, the KL term—and hence $I(\mathcal{D}; \theta)$—must grow. Decomposing

$$I(\mathcal{D}; \theta) = \underbrace{I(U^m; \theta)}_{\text{signal about true value}} + \underbrace{I(\mathcal{D}; \theta \mid U^m)}_{\text{residual: channel noise/bias}},$$

the useful signal $I(U^m; \theta)$ is capped by the capacity term in Proposition 3, so further optimization mainly increases the residual $I(\mathcal{D}; \theta \mid U^m)$ and pushes the model to fit structure in the feedback channel (rater quirks, reward-model artifacts, preferences for flattery and agreement) rather than in $U$ (Ngampruetikorn and Schwab, 2022). Empirically, this appears as sycophancy or reward hacking

| Dataset | High-capacity feedback (mean $u_{\text{total}}$) | Bottlenecked feedback (mean $u_{\text{total}}$) |
|---|---|---|
| Base (no DPO) | 40.88 | 40.88 |
| 50 pairs | 41.68 | 41.69 |
| 200 pairs | 41.75 | 41.53 |
| 500 pairs | 42.57 | 41.64 |
| 1000 pairs | **43.08** | **39.59** |

Table 1: Mean constitution scores $u_{\text{total}}$ under high-capacity vs. bottlenecked AI feedback as data size grows. Scores are measured by a separate oracle model that applies the five-rule constitution and sums per-rule integers in $[0, 10]$, so $u_{\text{total}} \in [0, 50]$; higher is better.

when feedback loss keeps dropping but spec-aware metrics plateau or degrade (Langosco et al., 2023; Sharma et al., 2024; Pan et al., 2022; Lin et al., 2024b). At that point, useful interventions are to temper optimization (smaller KL budgets, early stopping, explicit regularization on $I(\mathcal{D}; \theta)$) or to upgrade the feedback channel (clearer instructions, stronger rater models, richer labels), not simply to collect more data of the same type.

## 8 EMPIRICAL OBSERVATION OF CAPACITY LIMITS

**Setup.** We instantiate the feedback channel with AI feedback in a setting that mirrors a human-in-the-loop pipeline and we compare a high-capacity rater prompt with a bottlenecked prompt. Qwen2.5-3B-Instruct (Qwen et al., 2025) serves as the policy, and Qwen2.5-7B-Instruct serves as both oracle and rater, modulated only by its prompt. A five-rule constitution over truthfulness, harmlessness, instruction-following, clarity, and non-sycophancy defines a ground-truth target: for each prompt–answer pair the oracle applies all rules and returns per-rule scores in $\{0, \ldots, 10\}$ and their sum $u_{\text{total}} \in [0, 50]$, which we use only for evaluation. On the rater side this induces a concrete channel $U \to H \to Y$: in a high-capacity condition, the rater prompt restates the constitution and asks the model to choose, among several candidate responses to a prompt, the one that best satisfies all five rules; in a bottlenecked condition, the rater prompt emphasizes emotional validation, politeness, and agreement subject to a loose safety baseline, and chooses the most comforting candidate. In both cases we convert the best–worst decision into pairwise preferences $Y$ and train DPO policies (Rafailov et al., 2024b) on $m \in \{50, 200, 500, 1000\}$ preference pairs drawn from the same pool of prompts and candidate responses. All policies are evaluated by the same oracle on a held-out set of prompts using $u_{\text{total}}$, with averages and confidence intervals estimated over prompts and six random seeds. The same information-theoretic quantities apply if $H$ and $Y$ were produced by human raters through analogous interfaces instead of by a model.

**Results.** Table 1 reports mean oracle scores $u_{\text{total}}$ for the base 3B policy and the DPO policies under both feedback channels, and Figure 1 shows the same trends with 95% confidence intervals. The base model scores 40.88 out of a $[0, 50]$ scale. Under the high-capacity channel, mean $u_{\text{total}}$ improves roughly monotonically from 41.68 at 50 pairs to 43.08 at 1000 pairs. Under the bottlenecked channel, scores are essentially flat at small data sizes (41.69 at 50 pairs and 41.64 at 500 pairs) and drop sharply to 39.59 at 1000 pairs, below the base model.

**Analysis and Discussion.** The high-capacity condition is a setting where the feedback channel can transmit most of the five-rule constitution, so increasing the number of preference pairs converts more data into more information about $U$ and moves the policy closer to the oracle target until the curve saturates. In the bottlenecked condition only a small, tone-aligned slice of $U$ can pass through the compressed prompt, so the useful signal saturates quickly: small improvements at 50–500 pairs are followed by a clear degradation at 1000 pairs. This pattern matches the capacity picture in Sections 4 and 5: once $I(U; Y \mid S)$ has saturated under a low-capacity channel, further optimization mainly fits rater-specific shortcuts such as emotional validation and agreement, which the bottlenecked rater rewards but the constitution-based oracle penalizes. For practitioners, the experiment illustrates that scaling preference data under a narrow feedback interface can first help, then stall, and eventually

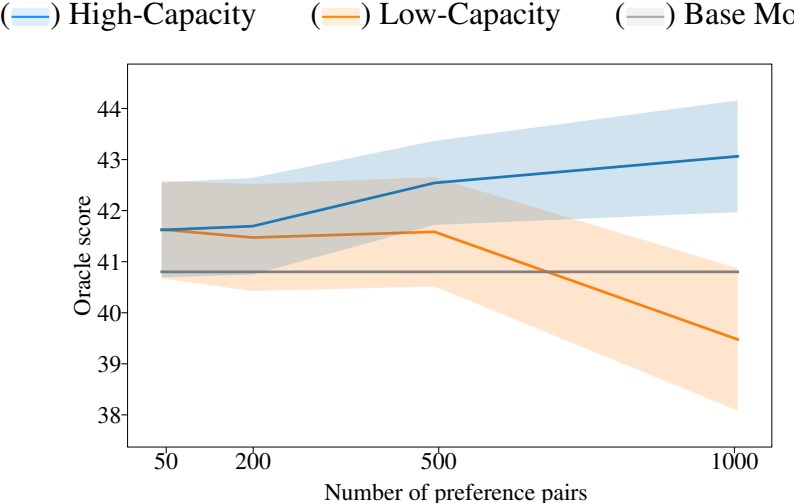

Figure 1: Capacity-limited alignment under high-capacity (blue) vs. bottlenecked (orange) AI feedback as the number of preference pairs increases. Curves show mean oracle constitution score $u_{\text{total}}$ over evaluation prompts and six random seeds, with 95% confidence intervals estimated from the empirical standard error. The horizontal gray line marks the base 3B policy with no DPO ($u_{\text{total}} = 40.88$). The vertical axis is the natural $[0, 50]$ scale of $u_{\text{total}}$, but the plot is zoomed to the range $[38, 44]$ for readability.

harm alignment to a richer specification, while strengthening the feedback channel through better prompts, more informative labels, or higher-capacity raters can keep gains from additional data aligned with the underlying value system.

## 9  CONCLUSION

Alignment is constrained by a finite information budget given by the capacity $\bar{C}_{\text{tot}|S}$ of the feedback channel from specification to labels. The central design choice is how to spend this budget across rules, tasks, prompts and raters, and how much capacity to allocate to high risk behavior. Model and algorithm changes matter only insofar as they help encode more of the latent target into parameters without amplifying rater shortcuts. A more quantitative practice of alignment will measure and allocate capacity explicitly, and will treat constitutions, interfaces and rater training as primary levers for increasing effective $\bar{C}_{\text{tot}|S}$ rather than simply enlarging data sets or KL budgets.

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

## A  NOTATION AND CROSS-REFERENCES

Throughout, $\log$ denotes the natural logarithm. Key references: single capacity inequality equation 7; Fano–packing converse Thm. 1; PAC–Bayes Thm. 2; expected KL identity Lemma 3; dataset information Lemma 4; capacity-control Proposition 3; Posterior Bayes–Loss Identity Lemma 7; Assumptions 2, 3; final statements in Sec. 6.

## B  LIMITATIONS AND PRACTICAL GUIDANCE

This appendix sketches how to check the main assumptions in concrete pipelines and when the bounds from equation 7, equation 9 and equation 25 should be read as tight or conservative. The loss–index link (Assumption 2) must be validated per task using the constructions in Appendix D.1. The capacity $\bar{C}_{\text{tot}|S}$ depends on the instantiated cognitive and articulation families $\mathcal{F}_{\text{cog}}$ and $\mathcal{F}_{\text{art}}$, and the residual information $\rho$ (Assumption 3) is algorithm-dependent.

### B.1  VALIDATING THE LOSS–INDEX LINK

Assumption 2 requires that mis-decoding an action as the wrong index $i$ implies loss at least $\varepsilon + \Delta$. For 0–1 classification and pairwise ranking this holds by construction (Appendix D.1); for truncated MSE it holds if prototypes form a separated packing and $\phi$ is a nearest-prototype rule. For structured or multi-objective losses, it should be treated as an empirical modeling condition: given $\{(u^{(i)}, a^{(i)})\}$ and $\phi$, sample actions near each $a^{(i)}$, score them with a strong evaluator for the underlying specification, and check that mis-decoded actions typically incur loss above the margin. When this only holds in a high-probability sense, the soft variant in Appendix M applies. If the loss is highly non-separable, effective $M$ is small and the Fano wall equation 9 becomes loose.

### B.2  OPERATIONALIZING CAPACITY IN PRACTICE

The quantity $\bar{C}_{\text{tot}|S}$ is defined via mutual information and the channel families $\mathcal{F}_{\text{cog}}$ and $\mathcal{F}_{\text{art}}$, which are not directly observed. A practical route is an oracle–rater calibration in which a concrete way of specifying desired behavior and a strong evaluator (oracle) for it are fixed and labels on the same held-out examples are collected from the production feedback channel. Agreement statistics or simple information-theoretic estimators of $I(U; Y \mid S)$ on this shared evaluation set then provide a task- and interface-specific lower bound on how much of the specification can traverse the loop. The cascade bound equation 7 is conservative because it uses a source-dependent minimum of per-stage suprema, so the theoretical wall in equation 9 may overestimate the information that actually reaches the learner. The aim is not to fix a particular numerical value, but to make explicit that whatever capacity is measured jointly constrains the achievable error floor and the learnable complexity.

### B.3  RESIDUAL INFORMATION AND ALGORITHMIC CHOICES

Assumption 3 isolates $\rho = I(\mathcal{D}; \theta \mid U^m)$ as the part of dataset–parameter mutual information that is not explained by the latent value sequence $U^m$. When training is nearly deterministic, long, and uses a very expressive model, $\rho$ can be large, reflecting overfitting to rater quirks or interface artifacts, and the capacity-explicit upper bound equation 25 becomes loose even if the lower bound is informative. Algorithmic choices such as injecting noise into updates, early stopping, or data-independent posterior compression (Appendix K) can keep $\rho$ finite and bring the PAC–Bayes ceiling closer to the channel limit. Reporting whether such mechanisms are used clarifies how much of the observed behavior is constrained by the feedback channel versus optimization details.

### B.4  WHEN ARE THE BOUNDS TIGHT OR LOOSE?

The Fano wall (Theorem 1) and the capacity-controlled PAC–Bayes ceiling equation 25 tend to be tightest when the codebook construction and the feedback channel both reflect the range of target configurations seen in deployment and operate near the source-dependent capacities $C_{\text{cog}|S}$ and $C_{\text{art}|S}$, and when the learning algorithm keeps $\rho$ moderate so that most of $I(\mathcal{D}; \theta)$ is spent on $U^m$ rather than noise. When these ingredients are missing, the bounds should be interpreted more qualitatively. A coarse or synthetic codebook together with a weak interface or rater pool can understate difficulty and make the wall pessimistic by inflating $\bar{C}_{\text{tot}|S}$ relative to realized $I(U; Y \mid S)$, while dataset shift can decouple $R_{\text{mix}}$ from deployed risk and a large value of $\rho$ can make the upper bound reflect optimization rather than channel structure. In such regimes, the framework is best used as a structural lens on feedback-limited alignment, with empirical diagnostics such as the high- versus low-capacity regimes in Sec. 7.3 guiding how to read the bounds.

## C  Conditional Capacities and the Cascade

**Proof of Proposition 1.** The cascade $U \to H \to Y$ given $S$ forms a Markov chain. The result follows directly from the data processing inequality (Shannon, 1948), which states that for such a chain, $I(U; Y \mid S = s) \leq I(U; H \mid S = s)$ and $I(U; Y \mid S = s) \leq I(H; Y \mid S = s)$. Taking suprema over the respective families yields $I(U; Y \mid S = s) \leq C_{\text{tot}|S}(s)$. Averaging over $S$ proves equation 7.

## D  Packing Constructions for Common Losses

### D.1  Separable Codebooks and the Loss–Index Link

We collect here the formal definition of separable codebooks and the loss–index link used in Section 4.1, together with remarks on concrete constructions.

**Definition 6** ($\Delta$-Separable Codebook). *A collection $\{(u^{(i)}, a^{(i)})\}_{i=1}^M \subset \mathcal{U} \times \mathcal{A}$ is called a $\Delta$-separable codebook for loss $\ell \in [0, 1]$ if*

$$\ell(u^{(i)}, a^{(i)}) \leq \varepsilon \quad \text{for all } i, \tag{16}$$

$$\ell(u^{(j)}, a^{(i)}) \geq \varepsilon + \Delta \quad \text{for all } j \neq i, \tag{17}$$

*for some $\varepsilon \in [0, 1 - \Delta]$. We write $\mathcal{C}(M, \Delta, \varepsilon)$ for the set of such codebooks.*

**Remark 1** (How to Build $\mathcal{C}(M, \Delta, \varepsilon)$ in Practice). *The construction of such codebooks (packings) is standard for minimax lower bounds in statistical decision theory and information theory (Shannon, 1948). For 0–1 classification, choose $a^{(i)}$ predicting class $i$, giving $\varepsilon = 0$ and $\Delta = 1$. For pairwise ranking with 0–1 pairwise loss averaged over all $\binom{n}{2}$ pairs, choose $a^{(i)}$ realizing ranking $i$, so $\varepsilon = 0$ and any two total orders differ on at least one pair, yielding $\Delta = 1/\binom{n}{2}$ after normalization to $[0, 1]$. For truncated MSE $\ell(u, a) = \min\{\|u - a\|^2/\tau^2, 1\}$, take $a^{(i)} = u^{(i)}$ on an $r$-separated packing of $\mathcal{U}$, then $\varepsilon = 0$ and it is consistent with Assump. 2 to use the common margin $\Delta = r^2/(4\tau^2)$ (the prototype cross-loss is $\geq r^2/\tau^2 \geq \Delta$, while the Voronoi misclassification loss is $\geq r^2/(4\tau^2)$). See the subsections below for concrete packings.*

**Assumption 2** (Loss–Index Link via a Measurable Partition). *Given a $\Delta$-separable codebook $\{(u^{(i)}, a^{(i)})\}_{i=1}^M$, there exists a measurable map $\phi : \mathcal{A} \times \mathcal{S} \to [M]$ (an "index decoder") such that for all $i$, all $s \in \mathcal{S}$, and all $a \in \mathcal{A}$,*

$$\phi(a, s) \neq i \implies \ell(u^{(i)}, a) \geq \varepsilon + \Delta. \tag{18}$$

**Remark 2** (When Assumption 2 Holds). *For 0–1 classification and pairwise ranking, let $\phi$ return the predicted class/ranking (allowing dependence on $S$ if needed); then equation 18 holds immediately. For truncated MSE, let $\phi$ be the nearest-prototype Voronoi partition under $\|\cdot\|$ (prototypes may depend on $S$). With an $r$-separated packing, any misclassification implies $\|a - u^{(i)}\| \geq r/2$, hence $\ell(u^{(i)}, a) \geq r^2/(4\tau^2)$, so equation 18 holds with the common choice $\Delta = r^2/(4\tau^2)$; equivalently, the separation condition is $r \geq 2\tau\sqrt{\Delta}$. See Appendix D for concrete packings.*

A soft high-probability variant of the loss–index link that yields a correspondingly slackened converse is provided in Appendix M.

## D.2 BINARY CLASSIFICATION

Let $\mathcal{U} = [M]$ and $\mathcal{A}$ the set of labels. Take $a^{(i)} = i$. Then $\varepsilon = 0$ and for any $j \neq i$, $\ell(u^{(j)}, a^{(i)}) = 1$, giving $\Delta = 1$. Let $\phi$ be the predicted label; Assump. 2 holds with margin 1.

## D.3 PAIRWISE RANKING WITH 0–1 LOSS

Let $u^{(i)}$ encode a total order over items and $\ell$ be the fraction of misordered pairs. Use $a^{(i)} = u^{(i)}$ (predict that order). Then $\varepsilon = 0$ and for any $j \neq i$, at least one pair flips, so $\Delta \geq 1/\binom{n}{2}$; with standard $\{0,1\}$ pairwise loss averaged over all $\binom{n}{2}$ pairs and normalized to $[0,1]$, the minimal separation is $\Delta = 1/\binom{n}{2}$. Let $\phi$ output the predicted order; Assump. 2 holds.

## D.4 TRUNCATED AND NORMALIZED MSE

Let $\ell(u,a) = \min\{\|u-a\|^2/\tau^2, 1\}$. Choose an $r$-separated packing $\{u^{(i)}\}_{i=1}^M$ in $\mathcal{U}$ (under $\|\cdot\|$), and set $a^{(i)} = u^{(i)}$. Then $\varepsilon = 0$ and, for $j \neq i$, the prototype cross-loss satisfies $\ell(u^{(j)}, a^{(i)}) \geq r^2/\tau^2$. To make Definition 6 and Assumption 2 hold with a single margin, take the common choice

$$\Delta = \frac{r^2}{4\tau^2}.$$

Indeed, for any $a$ misclassified by the nearest-prototype Voronoi rule, one has $\|a - u^{(i)}\| \geq r/2$, so $\ell(u^{(i)}, a) \geq r^2/(4\tau^2) = \Delta$. Since $r^2/\tau^2 \geq \Delta$, the prototype cross-loss condition in Definition 6 also holds.

## E FANO–PACKING CONVERSE DETAILS

**Lemma 1** (Risk ⇒ Index Error). *Under Assumption 2, for any decoder $\pi$ and $\phi$ as in equation 18, define $\hat{J} \triangleq \phi(\pi(Y,S), S)$. If $J$ is uniform over $[M]$ and $U = U^{(J)}$ is the codebook target (measurable in $(J,S)$), then*

$$\mathbb{E}\big[\ell(U, \pi(Y,S))\big] \geq (\varepsilon + \Delta)\,\mathbb{P}\{\hat{J} \neq J\}. \tag{19}$$

**Lemma 2** (Information Reduction: $J \to U \to Y$). *With $U = U^{(J)}$ measurable in $(J,S)$, we have the Markov chain $J \to U \to Y$ given $S$, and hence*

$$I(J;Y \mid S) \leq I(U;Y \mid S). \tag{20}$$

We expand the proof of Thm. 1. Let $J$ be uniform on $[M]$, $U = U^{(J)}$. With $\hat{J} = \phi(\pi(Y,S), S)$, Lemma 1 gives $R(\pi) \geq (\varepsilon + \Delta)\,\mathbb{P}\{\hat{J} \neq J\}$. The standard form of Fano's inequality (Shannon, 1948), when conditioned on $S$, implies that $H(J \mid Y,S) \leq \mathbb{P}\{\hat{J} \neq J\}\log(M-1) + h_2(\mathbb{P}\{\hat{J} \neq J\})$, which gives the more convenient bound

$$\mathbb{P}\{\hat{J} \neq J\} \geq 1 - \frac{I(J;Y \mid S) + \log 2}{\log M}.$$

Using $J \to U \to Y$ given $S$ (Lemma 2), we get equation 8; then apply equation 7 for equation 9.

### E.1 THE INFORMATION WALL

Define the information wall for a problem class $\mathcal{P}$ (set of admissible codebooks) by

$$\mathsf{Wall}\big(\bar{C}_{\mathrm{tot}|S}^{\mathrm{mix}}; \mathcal{P}\big) \triangleq \sup_{(M,\Delta,\varepsilon):\, \mathcal{C}(M,\Delta,\varepsilon)\in\mathcal{P}} (\varepsilon + \Delta)\left(1 - \frac{\bar{C}_{\mathrm{tot}|S}^{\mathrm{mix}} + \log 2}{\log M}\right)_+. \tag{21}$$

Here $\bar{C}_{\mathrm{tot}|S}^{\mathrm{mix}}$ is evaluated under the codebook-induced mixture distribution used in Theorem 1.

Then, for every decoder $\pi$,

$$\sup_{\mathcal{C}\in\mathcal{P}} R_{\mathrm{mix}}^{\mathcal{C}}(\pi) \geq \mathsf{Wall}(\bar{C}_{\mathrm{tot}|S}^{\mathrm{mix}}; \mathcal{P})$$

where $R_{\mathrm{mix}}^{(M,\Delta,\varepsilon)}(\pi)$ denotes the risk under the mixture induced by the chosen codebook (Bayes/minimax semantics from Thm. 1). Equivalently, this yields the standard minimax lower bound

$$\inf_{\pi} \sup_{\mathcal{C} \in \mathcal{P}} R_{\mathrm{mix}}^{\mathcal{C}}(\pi) \geq \mathsf{Wall}(\bar{C}_{\mathrm{tot}|S}^{\mathrm{mix}}; \mathcal{P}).$$

This replaces log-loss/posterior-entropy converses and is invariant to reparameterizations.

## F PAC–BAYES DETAILS AND RESIDUAL CONTROL

### F.1 KL DECOMPOSITION AND CAPACITY CONTROL

We collect the technical ingredients underlying the capacity-explicit PAC–Bayes bound summarized in Section 5.2.

**Lemma 3** (Expected KL Decomposition). *Fix a prior $Q$ that is independent of the dataset $\mathcal{D}$. Let $p(\theta)$ be the marginal of $\theta$ and $P(\cdot \mid \mathcal{D})$ be the posterior. Then*

$$\mathbb{E}_{\mathcal{D}}\big[\mathrm{KL}(P\|Q)\big] = I(\mathcal{D};\theta) + \mathrm{KL}\big(p(\theta)\,\|\,Q\big). \tag{22}$$

This identity underlies information-theoretic learning theory and expresses the expected KL term as the sum of the mutual information between data and parameters and a prior mismatch term (Xu and Raginsky, 2017; Russo and Zou, 2019). The mutual information piece is the one we will bound by the channel capacity.

**Lemma 4** (From $\mathcal{D}$ to $(U^m, S^m, Y^m)$). *Assume samples $(U_i, S_i)$ are i.i.d., and $Y_i$ are drawn conditionally independently via the human channel given $(U_i, S_i)$ as in Assumption 1. Let $\theta$ be any (possibly randomized) function of $\mathcal{D} \triangleq \{(Y_i, S_i)\}_{i=1}^m$. Then*

$$I(U^m;\theta) \leq I(U^m; Y^m, S^m) = I(U^m; S^m) + I(U^m; Y^m \mid S^m)$$

$$= \sum_{i=1}^m I(U_i; S_i) + \sum_{i=1}^m I(U_i; Y_i \mid S_i) = m\, I(U;S) + \sum_{i=1}^m I(U_i; Y_i \mid S_i). \tag{23}$$

**Proposition 3** (Capacity Control of $I(U^m;\theta)$). *Using equation 7 and Lemma 4,*

$$I(U^m;\theta) \leq m\, \bar{C}_{\mathrm{tot}|S} + m\, I(U;S). \tag{24}$$

**Convention.** *All mutual informations in this section are defined with respect to the underlying data-generating distribution (population quantities), and $\bar{C}_{\mathrm{tot}|S}$ is computed under the same source distribution; no averaging over the realized dataset is involved.*

### F.2 ALGORITHMIC RESIDUAL INFORMATION

**Assumption 3** (Residual Information of the Algorithm). *There exists $\rho \geq 0$ such that $I(\mathcal{D};\theta \mid U^m) \leq \rho$. It can be reduced by algorithmic noise (SGD temperature), early stopping, or posterior smoothing; see Appendix F. This term measures information about the particular sample beyond the latent value $U$ and parallels the "residual information" used in information-theoretic analyses of overfitting (Ngampruetikorn and Schwab, 2022).*

Practically, a data-independent randomized compression of the posterior enforces a finite residual, giving $\rho \leq \log K$ for any chosen codebook size $K$ without increasing $\mathrm{KL}(P\|Q)$ (see Appendix K). The idea of limiting information flow to improve generalization is widespread, though the causal link between compression and performance remains under debate (Kawaguchi et al., 2023; Saxe et al., 2019; Shwartz-Ziv et al., 2024; He et al., 2025). Here $\rho$ isolates information learned from $(Y, S)$ that is not about $U$, which is the target of such regularization.

**Corollary 2** (A Capacity-Aware Upper Bound). *Combining Lemma 3, Proposition 3, and Assumption 3, we have*

$$\mathbb{E}_{\mathcal{D}}\big[\mathrm{KL}(P\|Q)\big] \leq m\, \bar{C}_{\mathrm{tot}|S} + m\, I(U;S) + \rho + \mathrm{KL}\big(p(\theta)\,\|\,Q\big). \tag{25}$$

*Equation equation 25 controls the expectation of $\mathrm{KL}(P\|Q)$ over the draw of $\mathcal{D}$ and does not by itself yield a capacity-explicit high-probability bound. Appendix G gives a Markov-type lifting to high probability. Taking expectations in Thm. 2 and applying Jensen yields corresponding in-expectation variants.*

**Remark 3** (Conservative Interpretation). *When $\rho$ or $I(U;S)$ is large, capacity may not dominate the upper bound. Our statements should be read as: under Assumption 3 and moderate $I(U;S)$, both the converse (Thm. 1) and the PAC–Bayes upper bound are primarily driven by $\bar{C}_{\text{tot}|S}$.*

### F.3 Proofs of Lemma 3 and Lemma 4

**Lemma 3.** The identity is a foundational result in information-theoretic learning theory (Xu and Raginsky, 2017; Russo and Zou, 2019). The proof is as follows: with $Q$ independent of $\mathcal{D}$,
$$\mathbb{E}_{\mathcal{D}}[\text{KL}(P\|Q)] = \mathbb{E}_{\mathcal{D},\theta\sim P}\big[\log\frac{P(\theta|\mathcal{D})}{Q(\theta)}\big] = I(\mathcal{D};\theta) + \text{KL}(p(\theta)\|Q).$$

**Lemma 4.** Data processing gives $I(U^m;\theta) \leq I(U^m;\mathcal{D})$. Then $I(U^m;Y^m,S^m) = I(U^m;S^m) + I(U^m;Y^m \mid S^m)$, with $I(U^m;S^m) = \sum_i I(U_i;S_i) = mI(U;S)$ by i.i.d. For the conditional term, under the i.i.d. source and the memoryless channel $p(y_i \mid u_i, s_i)$, we have $p(u^m \mid s^m) = \prod_i p(u_i \mid s_i)$ and hence $p(y^m \mid s^m) = \prod_i \int p(y_i \mid u_i, s_i) p(u_i \mid s_i) du_i = \prod_i p(y_i \mid s_i)$. Therefore $H(Y^m \mid S^m) = \sum_i H(Y_i \mid S_i)$ and $H(Y^m \mid U^m, S^m) = \sum_i H(Y_i \mid U_i, S_i)$, which gives $I(U^m;Y^m \mid S^m) = \sum_i I(U_i;Y_i \mid S_i)$.

### F.4 Controlling the Residual Term

We list standard mechanisms to enforce Assump. 3. These methods all serve to regularize the information that the learned parameters $\theta$ contain about the specific training dataset $\mathcal{D}$. Algorithmic noise can be introduced by injecting Gaussian noise into updates or by using high-temperature posteriors; early stopping bounds the mutual information by limiting the number of optimization steps; posterior smoothing mixes the learned posterior with the prior. The general goal of controlling information flow, often framed as a form of compression, is a central theme in understanding deep learning generalization, although its precise role and benefits are still actively debated (Kawaguchi et al., 2023; Saxe et al., 2019; Shwartz-Ziv et al., 2024; He et al., 2025).

## G From Expectation to High Probability

This section provides a simple method to convert our expectation-based capacity bound on the KL-divergence into a high-probability statement. This type of conversion from expectation to high-probability bounds is a common step in applying learning-theoretic results. More sophisticated techniques can yield tighter, anytime-valid bounds that hold uniformly over time (Rodríguez-Gálvez et al., 2024). A direct application of Markov's inequality suffices.

**Lemma 5** (Markov Lift for the KL Term). *Let $X \triangleq \text{KL}(P\|Q) \geq 0$ denote the (dataset-dependent) PAC–Bayes KL term. For any $\eta \in (0,1)$, with probability at least $1 - \eta$ (over the draw of $\mathcal{D}$),*
$$X \leq \frac{\mathbb{E}_{\mathcal{D}}[X]}{\eta}.$$

*Proof.* Since $X \geq 0$ and $\mathbb{E}_{\mathcal{D}}[X] < \infty$ under the conditions of Theorem 2, Markov's inequality gives
$$\mathbb{P}\bigg\{ X > \frac{\mathbb{E}_{\mathcal{D}}[X]}{\eta} \bigg\} \leq \eta.$$
Equivalently, with probability at least $1 - \eta$ we have $X \leq \mathbb{E}_{\mathcal{D}}[X]/\eta$, as claimed. $\square$

**Corollary 3** (A Capacity-Aware High-Probability Upper Bound). *Fix $\delta, \eta \in (0,1)$. With probability at least $1 - \delta - \eta$ (over the draw of $\mathcal{D}$), the PAC–Bayes bound of Thm. 2 implies*
$$\mathbb{E}_{\theta\sim P}\big[R_{\text{obs}}(\theta)\big] \leq \mathbb{E}_{\theta\sim P}\big[\widehat{R}_m^{\text{obs}}(\theta)\big] + \sqrt{\frac{\mathbb{E}_{\mathcal{D}}[\text{KL}(P\|Q)]/\eta + \log(1/\delta)}{2m}}.$$
*Combining with Cor. 2 and applying a union bound yields, with the same probability,*
$$\mathbb{E}_{\theta\sim P}\big[R_{\text{obs}}(\theta)\big] \leq \mathbb{E}_{\theta\sim P}\big[\widehat{R}_m^{\text{obs}}(\theta)\big] + \sqrt{\frac{m\bar{C}_{\text{tot}|S} + m\,I(U;S) + \rho + \text{KL}(p(\theta)\|Q)}{2m\,\eta}} + \frac{\log(1/\delta)}{2m}.$$

*All constants are explicit; the price of eliminating the dataset randomness in $\text{KL}(P\|Q)$ is the slack parameter $\eta$.*

## H  Loss Truncation and Normalization

For unbounded losses such as MSE, define $\ell(u,a) = \min\{\|u-a\|^2/\tau^2, 1\}$ for a scale $\tau > 0$ (report $\tau$ when plotting). All PAC–Bayes statements and Thm. 1 require only $\ell \in [0,1]$; truncation ensures this and keeps statements coordinate-free.

## I  Loss–Observable Link and Risk Transfer

**Assumption 4** (Loss–Observable Link). *There exist constants $\alpha \geq 0$ and $\beta \geq 0$ such that for all measurable actions $a \in \mathcal{A}$ and all $(y,s)$ in the support of $(Y,S)$,*

$$\mathbb{E}\big[\ell(U,a) \,\big|\, Y = y,\, S = s\big] \ \leq\ \alpha\, \tilde{\ell}(y,s,a) \ +\ \beta\,. \tag{26}$$

**Lemma 6** (Risk Transfer). *Under Assumption 4, for any (possibly randomized) decoder $\pi_\theta$,*

$$\mathbb{E}\big[\ell\big(U, \pi_\theta(Y,S)\big)\big] \ \leq\ \alpha\, \mathbb{E}\big[\tilde{\ell}\big(Y, S, \pi_\theta(Y,S)\big)\big] \ +\ \beta\,. \tag{27}$$

*In particular, when $\mathcal{D}$ is drawn from the same codebook-induced mixture distribution used in Theorem 1 and we take $\theta \sim P(\cdot \mid \mathcal{D})$ and expectation over both $\theta$ and the data, we have*

$$\mathbb{E}_{\theta \sim P}\big[R_{\mathrm{mix}}(\pi_\theta)\big] \ \leq\ \alpha\, \mathbb{E}_{\theta \sim P}\big[R_{\mathrm{obs}}(\theta)\big] \ +\ \beta\,.$$

Under the Loss–Observable Link (Assumption 4) and when $\mathcal{D}$ is drawn from the same codebook-induced mixture used in Thm. 1, combining Lemma 6 with the PAC–Bayes bound equation 12 yields the following direct upper bound on the true risk:

$$\mathbb{E}_{\theta \sim P}\big[R_{\mathrm{mix}}(\pi_\theta)\big] \ \leq\ \alpha\Big(\mathbb{E}_{\theta \sim P}\big[\widehat{R}_m^{\mathrm{obs}}(\theta)\big] \ +\ \sqrt{\tfrac{\mathrm{KL}(P\|Q)+\log(1/\delta)}{2m}}\Big) \ +\ \beta\,.$$

Together with equation 11, this yields two-sided bounds on the same quantity $R_{\mathrm{mix}}$, with explicit constants $(\alpha, \beta)$ coming from the link assumption.

Finally, if the dataset $\mathcal{D}$ is drawn from the same codebook-induced mixture as in Theorem 1 and we take the canonical observable loss $\tilde{\ell} = \tilde{\ell}^\star$ (Appendix J), then $R_{\mathrm{obs}}(\theta) = R_{\mathrm{mix}}(\pi_\theta)$ and Eq. equation 12 becomes a high-probability upper bound on the same risk $R_{\mathrm{mix}}$ as in the converse; together with Eq. equation 11, this gives a two-sided bound without additional link assumptions. In alignment terms, for any feedback alignment scheme that can be summarized by a codebook and a feedback channel, the same scalar quantity $\bar{C}_{\mathrm{tot}|S}$ is the main driver of both the floor (through the Fano wall) and the ceiling (through the PAC-Bayes KL budget) on the achievable true alignment error under that scheme.

## J  Posterior Bayes–Loss Identity

**Lemma 7** (Posterior Bayes–Loss Identity). *Fix any bounded loss $\ell \in [0,1]$ and define $\tilde{\ell}^\star(y,s,a) \triangleq \mathbb{E}[\ell(U,a) \mid Y = y,\, S = s]$. Then for any (possibly randomized) decoder $\pi_\theta$ and any data distribution over $(U, S, Y)$,*

$$\mathbb{E}\big[\tilde{\ell}^\star(Y, S, \pi_\theta(Y,S))\big] \ =\ \mathbb{E}\big[\ell(U, \pi_\theta(Y,S))\big]\,.$$

*Proof.* By the tower property of conditional expectation, $\mathbb{E}[\tilde{\ell}^\star(Y, S, \pi_\theta(Y,S))] = \mathbb{E}\{\mathbb{E}[\ell(U, \pi_\theta(Y,S)) \mid Y,S]\} = \mathbb{E}[\ell(U, \pi_\theta(Y,S))]$. $\qquad\square$

## K  Residual Control via Posterior Compression

Let $\theta \sim P(\cdot \mid \mathcal{D})$ be the (possibly randomized) learner parameter. Let $W$ be an auxiliary random seed, independent of $(U^m, S^m, Y^m)$. Consider a data-independent randomized quantizer $T$ that maps $\theta$ to $\tilde{\theta} = T(\theta, W)$ taking at most $K$ distinct values. Let $P_c$ and $Q_c$ be the pushforwards of $P$ and $Q$ through $T$. Then:

**Lemma 8** (Residual control by compression). $I(\mathcal{D}; \tilde{\theta} \mid U^m) \ \leq\ H(\tilde{\theta}) \ \leq\ \log K$.

*Proof.* $I(\mathcal{D}; \tilde{\theta} \mid U^m) \le H(\tilde{\theta})$, and $H(\tilde{\theta}) \le \log K$ since $\tilde{\theta}$ takes at most $K$ values. $\qquad\square$

**Lemma 9** (KL does not increase under post-processing). $\mathrm{KL}(P_c \| Q_c) \le \mathrm{KL}(P \| Q)$.

Using $P_c, Q_c$ in Theorem 2 and Lemma 3 yields the capacity-aware bound of Corollary 2 with $\rho \le \log K$. This approach is conceptually related to other works that leverage model compression or selection of small representative subsets to derive non-vacuous generalization bounds for overparameterized models (Leblanc et al., 2025; Lotfi et al., 2024).

## L  CONTEXT COARSENING

**Remark 4** (Source-dependent min and achievability). *Definitions 3–5 use per-stage feasibility (Assumption 1), so equation 6 is an immediate consequence of data processing. The quantities $C_{\mathrm{cog}|S}(s)$ and $C_{\mathrm{art}|S}(s)$ are computed under the actual source $P(U \mid S = s)$, not maximized over all inputs, and are therefore analogous to rate–distortion quantities for a fixed source (Shannon, 1959). In general $\min\{\sup I(U; H \mid S = s), \sup I(H; Y \mid S = s)\}$ is only an upper bound for the cascade because the per-stage optimizers may be incompatible, so equality should not be expected.*

Let $S' = T(S)$ for a measurable (data-release) channel $T$ such that $U \to S \to S'$ forms a Markov chain (that is, $S'$ is generated from $S$ without direct access to $U$). Then by data processing $I(U; S') \le I(U; S)$. All definitions and bounds in the paper hold verbatim with $S'$ in place of $S$, with capacities recomputed as $\bar{C}_{\mathrm{tot}|S'}$ and dataset information term $m\, I(U; S')$ replacing $m\, I(U; S)$. Thus, for any preprocessor $T$, Corollary 2 becomes

$$\mathbb{E}_{\mathcal{D}}\big[\mathrm{KL}(P \| Q)\big] \le m\, \bar{C}_{\mathrm{tot}|S'} + m\, I(U; S') + \rho + \mathrm{KL}(p(\theta) \| Q).$$

Choosing $T$ to enforce $I(U; S') \le \kappa$ makes the interference term $m\, \kappa$ explicit.

## M  SOFT LOSS–INDEX LINK

Assume there exists a measurable $\phi : \mathcal{A} \times \mathcal{S} \to [M]$ and parameters $\varepsilon, \Delta \ge 0$, $\zeta \in [0, 1)$ such that for all $i$ and all $a$,

$$\mathbb{P}\big\{ \mathbb{E}[\ell(U^{(i)}, a) \mid S] \ge \varepsilon + \Delta \,\big|\, \phi(a, S) \ne i \big\} \ge 1 - \zeta.$$

Then for any decoder $\pi$ with $\hat{J} = \phi(\pi(Y, S), S)$, writing $E \triangleq \{\hat{J} \ne J\}$ and $G \triangleq \big\{\mathbb{E}[\ell(U^{(J)}, \pi(Y, S)) \mid S] \ge \varepsilon + \Delta\big\}$, we have

$$\mathbb{E}\big[\ell(U, \pi(Y, S))\big] = \mathbb{E}\Big[\mathbb{E}\big[\ell(U^{(J)}, \pi(Y, S)) \mid S\big] \cdot \mathbf{1}_E\Big] + \mathbb{E}\Big[\mathbb{E}\big[\ell(U^{(J)}, \pi(Y, S)) \mid S\big] \cdot \mathbf{1}_{E^c}\Big]$$
$$\ge (\varepsilon + \Delta)\, \mathbb{P}(E \cap G),$$

hence

$$\mathbb{E}\big[\ell(U, \pi(Y, S))\big] \ge (\varepsilon + \Delta)\, \mathbb{P}\{\hat{J} \ne J\} - (\varepsilon + \Delta)\, \mathbb{P}(E \cap G^c).$$

By the assumption, $\mathbb{P}(G^c \mid E) \le \zeta$, so $\mathbb{P}(E \cap G^c) \le \zeta\, \mathbb{P}(E) \le \zeta$ and consequently

$$\mathbb{E}\big[\ell(U, \pi(Y, S))\big] \ge (\varepsilon + \Delta)\, \mathbb{P}\{\hat{J} \ne J\} - \zeta.$$

A slightly tighter but equivalent multiplicative form also holds:

$$\mathbb{E}\big[\ell(U, \pi(Y, S))\big] \ge (\varepsilon + \Delta)\,(1 - \zeta)\, \mathbb{P}\{\hat{J} \ne J\}.$$

Consequently, Theorem 1 holds with an additive $-\zeta$ (or multiplicative $(1 - zeta)$) slack in the lower bound.

## N  USE OF LARGE LANGUAGE MODELS

The author utilized Large Language Models as assistive tools in preparing this manuscript. Their applications included literature discovery, language refinement, and the formal review of mathematical derivations. The author directed the entire process and takes full responsibility for the final content and the accuracy of all theoretical claims.

