# OpenReview forum: "The Alignment Bottleneck"
_ICLR.cc/2026/Conference — Submitted to ICLR 2026_

### Official Review · Reviewer_y4Pn · 2025-10-31

**Soundness:** 2
**Presentation:** 1
**Contribution:** 1
**Rating:** 2
**Confidence:** 2

**Summary:**

This paper analyzes the AI alignment problem from an information-bottleneck perspective. The process is modeled as a two-stage cascade, modeling the users’ true preferences ($U$), the noisy information they emit ($Y$) through and intermediate stage ($H$), and the context ($S$). The authors present a Fano risk lower bound under mixture assumptions, a PAC-Bayes upper bound, and discuss the theoretical implications of the results.

**Strengths:**

* Interesting perspective on a timely and well-motivated topic.
* Analysis provides both lower and upper bounds.

**Weaknesses:**

* Significance of results is unclear, and practical implications are not explicitly discussed.
* Presentation is unclear, and the paper was slightly hard to follow.
* No empirical validation of findings.

**Questions:**

* What are the channel capacities expected to appear in practice?
* What are the practical implications of the presented results, and how may they inform the design of AI systems in practice?
* What are the limitations of the analysis? When is it expected to hold "tightly", and when are assumptions expected to break?

---

> ### Author Response · Authors · 2025-11-26
> **Response to Reviewer y4Pn (1/2)**
>
> Thank you for the candid review and for pointing out clearly where the current draft is not yet useful enough for people working on feedback alignment with humans or models.
>
> **Weakness 1: Significance of results is unclear, and practical implications are not explicitly discussed.**
>
> Our main practical message is that, for a fixed feedback setup, there is a hard feedback budget, so beyond a certain point more feedback data or stronger optimization cannot reliably reduce misalignment under a given safety rubric and can even make sycophancy worse. In the revision we keep the theoretical results and, as outlined in the global comments **Primary Contribution: The Capacity Coupled Structure** and **Revision Plan**, we will explain in plain language when additional feedback should help, when it is expected to hurt by overfitting to rater shortcuts, and when one should instead change the feedback interface or simplify the rubric.
>
> **Weakness 2: Presentation is unclear, and the paper was slightly hard to follow.**
>
> As described in the global comment **Revision Plan**, we will keep the technical content but explicitly tie the notation to a concrete picture with a constitution or rubric, an underlying outcome that is correct according to that rubric, a rater with limited attention, and the logged label, and we will add explanations after each main result that state what it predicts for a real feedback aligned system, following the mapping in **Operationalizing U as Constitutions and Rubrics** and **Bounded Rationality as the Basis for Channel Limits**.
>
> **Weakness 3: No empirical validation of findings.**
>
> We have added a controlled synthetic preference training experiment with Qwen models that shows the predicted capacity wall: under a high capacity rater prompt that enforces all five rules of the constitution, more data steadily improves the oracle score, while under a bottlenecked rater prompt that mainly uses tone and agreeableness, performance saturates quickly and then falls below the base model as the policy becomes more sycophantic. A concise version of this experiment, including the results table in the global comment **Empirical Observation of Capacity Limits**, **has been integrated into the revised paper as Section 8**, and we will release the code and synthetic data.

---

> ### Author Response · Authors · 2025-11-26
> **Response to Reviewer y4Pn (2/2)**
>
> **Question 1: What are the channel capacities expected to appear in practice?**
>
> In practical feedback alignment pipelines that rely on binary preferences or short rating scales under tight annotation budgets, each label provides only a limited, coarse grained signal about how well the output follows the underlying rubric, rather than a detailed assessment of every rule. In the revision we explain that, for a given pipeline, one can approximate the effective capacity of the feedback process by fixing a concrete constitution, collecting oracle judgments according to that constitution together with human or model rater outputs on the same data, and checking how well the oracle’s decisions can be recovered from the logged labels, as discussed in **Operationalizing U as Constitutions and Rubrics**.
>
> **Question 2: What are the practical implications of the presented results, and how may they inform the design of AI systems in practice?**
>
> Practically, our results say that if you try to teach a complex safety rubric through a very cheap feedback channel such as single clicks or coarse scores, then beyond some point more of the same feedback or stronger optimization mainly teaches the system to exploit rater shortcuts instead of the underlying safety rules. In our synthetic experiment this appears as a policy that becomes more agreeable and flattery seeking while its score under the full constitution drops below the base model. In the revision we make the design guidance explicit: when you see signs of this regime, you should either simplify what you are trying to teach in that dataset or enrich the feedback, for example with multi aspect labels, structured rubrics, textual critiques, more time per item or more raters or model raters per item, and you should choose optimization strength and KL budgets in line with the reliability of this feedback channel.
>
> **Question 3: What are the limitations of the analysis? When is it expected to hold "tightly", and when are assumptions expected to break?**
>
> The analysis is aimed at feedback alignment pipelines, including settings with human raters and settings with AI raters such as RLAIF, where there is a reasonably fixed written constitution or rubric, one shot labels, and limited attention per judgment, and it focuses on how much of the intended rubric signal can realistically make it into the logged labels under those conditions. We expect the conclusions about a feedback driven floor on alignment and the risk of channel overfitting to be most accurate when there is a stable safety policy, feedback is short and local, and bounded human or model attention is the main bottleneck; they become less accurate when oversight is highly interactive or multi stage, when the target behaviour is very open ended and evolving rather than approximated by a written rubric, or when other reward sources dominate the effect of the feedback channel. In the revision we state these caveats explicitly and present the framework as a structural lens on feedback limited alignment rather than a full model of all alignment schemes.

---

> > ### Comment · Reviewer_y4Pn · 2025-11-27
> >
> > Thank you for the detailed response! After reviewing the rebuttal and the other reviews, my main concerns remain. While the authors express their commitment to improve clarity and expand the discussion of applicability, the proposed steps are still not sufficiently concrete for me to assess their impact, and I am concerned that the scope of the proposed revisions will require additional review. That being said, I also acknowledge that I may be evaluating the work from a perspective outside the core information theory and alignment communities, and that the target audience may differ. Given this, I am maintaining my original score (reject with low confidence).

---

> > > ### Author Response · Authors · 2025-12-03
> > > **Thank you for your feedback**
> > >
> > > We thank Reviewer y4Pn for the candid feedback and for acknowledging the different perspectives on this topic. We understand your initial concern regarding the concreteness of the proposed revisions. To address this, we have fully implemented the experimental validation and the practical mapping into the newly uploaded PDF, moving from proposed plans to concrete results. We appreciate your time spent reviewing our work.

---

### Official Review · Reviewer_q9VB · 2025-11-01

**Soundness:** 3
**Presentation:** 3
**Contribution:** 3
**Rating:** 6
**Confidence:** 3

**Summary:**

The paper models human--AI feedback as a two-stage, capacity-limited channel (U - H - Y) conditioned on context \(S\). It derives (i) a data-size--independent Fano-style lower bound on true risk using separable codebooks, and (ii) a PAC--Bayes upper bound whose KL term is explicitly controlled by the same channel capacity via $C_{tot|S}$ Under a canonical observable loss and matched codebook mixture, these yield a two-sided ``Alignment Performance Interval,'' implying that more labels alone cannot beat the lower wall; required capacity scales with value complexity ${\log}M$; and over-optimization fits residual channel regularities (e.g., sycophancy or reward hacking).

**Strengths:**

1 Clear, simple formalization of the human loop. The cascade $U-H-Y$ with cognitive and articulation capacities is crisply defined and linked to information-bottleneck/rate--distortion intuitions; a central proposition bounds  $I(U;Y|S)$ by the average total capacity.

2. The key novelty is turning the KL complexity in PAC--Bayes into an environmental budget, aligning the ceiling with the same capacity that drives the Fano floor---an elegant, unifying perspective.

3. The consequences of the paper are (i) a lower bound independent of dataset size, (ii) necessary capacity scaling with  ${log}M$, and (iii) a mechanism for channel overfitting---translate theory into design levers (measure/allocate capacity, manage value complexity, regularize residual information).

**Weaknesses:**

No empirical validation: The theory is compelling, but there is no empirical study (even toy) quantifying $C_{tot|s}$ or demonstrating the predicted saturation/overfitting behavior under controlled capacity budgets across alignment protocols.

**Questions:**

My only question is regarding the empirical evaluations. Do the authors have any plans to get any empirical results?

---

> ### Author Response · Authors · 2025-11-26
> **Response to Reviewer q9VB**
>
> We are grateful to Reviewer q9VB for the positive assessment and for highlighting the importance of empirical validation.
>
> Empirical validation (Weakness + Question).
>
> As described in our global comment Empirical Observation of Capacity Limits, we have conducted a controlled synthetic DPO experiment that directly targets the saturation / overfitting behavior you asked about. Using Qwen2.5-3B as the policy and Qwen2.5-7B as both Oracle $U$ and rater $H$ under a 5-rule constitution, we compare:
>
> - a high-capacity channel ($H_{\text{full}}$) whose prompt enforces all five rules, and
>
> - a bottlenecked channel ($H_{\text{limited}}$) whose prompt collapses evaluation to cheap heuristics (agreeableness/tone) and weakens truthfulness checks.
>
>
> As we increase the dataset size, Oracle scores under $H_{\text{full}}$ improve roughly monotonically, while under $H_{\text{limited}}$ they follow the predicted pattern of **saturation followed by degradation**: scores are essentially flat at small data sizes ($50$--$500$ pairs) and finally drop below the base model accompanied by more sycophantic behavior.
>
> As outlined in our global comment **Revision Plan**, **we have (1) integrated** a concise version of this experiment into the main text (**Section 8**), explicitly linking it to the “alignment performance interval” and the capacity wall, and (2) we will provide the full code and synthetic data details upon release.

---

### Official Review · Reviewer_2NMe · 2025-11-02

**Soundness:** 3
**Presentation:** 3
**Contribution:** 3
**Rating:** 6
**Confidence:** 3

**Summary:**

This paper argues that aligning LLMs with human values is limited by human cognitive capacity. The authors treat human feedback as a bounded-information channel and show that there is a hard lower bound on how well alignment can work when human judgment is limited. They combine classical information-theoretic tools (Fano bounds) and PAC-Bayes theory to prove that even with more data, alignment performance cannot surpass this capacity bottleneck.

**Strengths:**

The paper is mathematically rigorous and formally couple a Fano lower bound with a PAC-Bayes upper bound using the same human-feedback capacity term, yielding an alignment performance interval that clarifies when and why increasing data does not improve alignment.

**Weaknesses:**

-Related prior work around the connection between bounded rationality and alignment is missing. For instance, see Bounded Rationality for LLMs: Satisficing Alignment at Inference-Time (arXiv:2505.23729), which appears to have explicitly connect alignment and bounded rationality. Incorporating this reference would help contextualize your contribution and clarify how your information-theoretic perspective differs from and extends that prior framing.

- But as such, the connection made between bounded rationality and information theory seems forced, without much discussion or evidence. While the theoretical analysis is well done, the need for it and the key rationale behind studying this problem is not clear.

- The authors immediately went into formulating feedback loop as two stage cascade framework, but how it would help or what problem it is aiming to solve exactly is unclear.

- In problem setup, what is the exact bottleneck of alignment is not defined, or clear from the discussion.

- Is the analysis presented in the paper is for parametrized settings or it does not matter, a discussion would help?

- Would the analysis of this work could shed some light or guidance on the type of feedback one should use for alignment, such as preference feedback?  Is it optimal?

- There are no empirical results in the paper. I understand that the work in theoretical in nature, but is it possible to provide some basic experiments to may be just connect what is the lower bound, and how we are doing currently with the available methods to motivate the importance of these lower bounds.

Note: I will rely on other reviewers to comment on the mathematical novelty of this work, since I am not an expert in information theory.

**Questions:**

Please refer to the discussion in weaknesses.

---

> ### Author Response · Authors · 2025-11-26
> **Response to Reviewer 2NMe (1/3)**
>
> We thank Reviewer 2NMe for the thoughtful and constructive review and the positive overall assessment.
>
> **Notation reminder.** For clarity, in our notation, $U$ denotes the rubric/constitution configuration, $H$ the rater’s internal judgment under bounded cognition, and $Y$ the observed feedback (e.g., preference or score). We keep this notation throughout and avoid redefining it in each answer.
>
> **Weakness 1: Related prior work around the connection between bounded rationality and alignment is missing. For instance, see Bounded Rationality for LLMs: Satisficing Alignment at Inference-Time (arXiv:2505.23729), which appears to have explicitly connect alignment and bounded rationality. Incorporating this reference would help contextualize your contribution and clarify how your information-theoretic perspective differs from and extends that prior framing.**
>
> We are grateful to Reviewer 2NMe for pointing us to _Bounded Rationality for LLMs: Satisficing Alignment at Inference-Time_ [1], which indeed directly connects bounded rationality and LLM alignment. We will add and discuss this work in the revision. Conceptually, [1] starts from Simon’s satisficing view and proposes SITAlign, an _inference-time_ alignment algorithm that maximizes a primary reward subject to threshold constraints on secondary rewards, together with suboptimality bounds and empirical evaluations. Our paper instead takes an _algorithm-agnostic_ perspective: we model human feedback as a bounded-capacity cognitive channel and show that this capacity term simultaneously controls a Fano-type lower bound and a PAC-Bayes upper bound on alignment error. Thus, while [1] analyzes a particular satisficing decoding scheme under bounded rationality, our results characterize the **fundamental information-theoretic limits** that apply to any alignment method that relies on bounded-rational human judgment.
>
> Beyond [1], we will also situate our work within the broader literature that explicitly links bounded/resource rationality and alignment, including alignment with unreliable supervision [2], resource-rational contractualist frameworks [3], critiques of preference-based alignment that emphasize resource-limited reasoning [4], and free-energy–based models of interactions between boundedly-rational agents with alignment motivations [5]. We will briefly review these works in a new subsection “Bounded rationality and AI alignment” and clarify that they are primarily algorithmic or normative frameworks, whereas our contribution is a unified information-theoretic treatment of how bounded human cognitive capacity induces an inherent **alignment bottleneck**.
>
> $\textbf{References:}$
>
> [1] Mohamad Chehade, Soumya Suvra Ghosal, Souradip Chakraborty, Avinash Reddy, Dinesh Manocha, Hao Zhu, and Amrit Singh Bedi. _Bounded Rationality for LLMs: Satisficing Alignment at Inference-Time._ ICML 2025.
>
> [2] Shivam Singhal, Cassidy Laidlaw, and Anca Dragan. _Achieving AI Alignment with Unreliable Supervision._ UC Berkeley EECS Technical Report UCB/EECS-2024-148, 2024.
>
> [3] Sydney Levine, Matija Franklin, Tan Zhi-Xuan, Secil Yanik Guyot, Lionel Wong, Daniel Kilov, Yejin Choi, Joshua B. Tenenbaum, Noah Goodman, Seth Lazar, and Iason Gabriel. _Resource Rational Contractualism Should Guide AI Alignment._ Preprint, 2025.
>
> [4] Tan Zhi-Xuan, Micah Carroll, Matija Franklin, and Hal Ashton. _Beyond Preferences in AI Alignment._ arXiv:2408.16984, 2024.
>
> [5] David Hyland, Tomáš Gavenčiak, Lancelot Da Costa, Conor Heins, Vojtech Kovarik, Julian Gutierrez, Michael J. Wooldridge, and Jan Kulveit. _Free-Energy Equilibria: Toward a Theory of Interactions Between Boundedly-Rational Agents._ ICML Workshop on Models of Human Feedback for AI Alignment, 2024.

---

> ### Author Response · Authors · 2025-11-26
> **Response to Reviewer 2NMe (2/3)**
>
> **Weakness 2: But as such, the connection made between bounded rationality and information theory seems forced, without much discussion or evidence. While the theoretical analysis is well done, the need for it and the key rationale behind studying this problem is not clear.**
>
> We kindly refer the reviewer to our global comment **Bounded Rationality as the Basis for Channel Limits**, where we explain in detail why bounded rationality naturally leads to a finite-capacity channel model. In short, cognitive science and decision theory consistently model humans as **resource-limited** agents whose judgments are lossy compressions of rich underlying values—formally captured via information constraints and rate–distortion tradeoffs [1–3]. Building on this established line, we treat human feedback in alignment as passing through a cognitively constrained internal representation, so that “capacity” is precisely a quantitative measure of how much value-relevant information a boundedly rational rater can transmit. Our information-theoretic bounds are thus not an arbitrary mathematical choice, but the **direct consequence** of modeling human raters in the same information-limited way that prior work models human and artificial decision-makers; the goal of the paper is to spell out what this widely accepted cognitive limitation implies for the ultimate limits of alignment.
>
> $\textbf{References:}$
>
> [1] Falk Lieder and Thomas L. Griffiths. _Resource-rational analysis: Understanding human cognition as the optimal use of limited computational resources._ Behavioral and Brain Sciences, 43:e1, 2020.
>
> [2] Dilip Arumugam and Benjamin Van Roy. _Deciding What to Learn: A Rate-Distortion Approach._ International Conference on Machine Learning (ICML), 2021.
>
> [3] Dilip Arumugam, Mark K. Ho, Noah D. Goodman, and Benjamin Van Roy. _Bayesian Reinforcement Learning with Limited Cognitive Load._ Open Mind, 8, 395–438, 2024.
>
> **Weakness 3: The authors immediately went into formulating feedback loop as two stage cascade framework, but how it would help or what problem it is aiming to solve exactly is unclear.**
>
> As detailed in our global comment **Operationalizing $U$ as Constitutions and Rubrics**, the two-stage framework is introduced to answer a concrete question: **in a realistic human-in-the-loop pipeline, where exactly is value information about $U$ lost, and which part of the process is the true alignment bottleneck?** Splitting the channel into cognitive compression $U \to H$ and articulation $H \to Y$ lets us define separate capacities and then bound the total information $I(U;Y\mid S)$ that any learner can access. This is what allows us to couple a Fano-type lower bound and a PAC–Bayes upper bound via the same capacity term. In the revision, we will add a one-sentence explanation in Sec. 3 stating this motivation explicitly.
>
> **Weakness 4: In problem setup, what is the exact bottleneck of alignment is not defined, or clear from the discussion.**
>
> In our analysis, the **alignment bottleneck** is the effective information constraint of the feedback channel, summarized by the capacity term _total capacity_. This term upper-bounds the amount of information about the value configuration $U$ that can reach the feedback $Y$, and it appears on both sides of our main results: (i) in the Fano-type lower bound on true risk and (ii) in the PAC–Bayes upper bound on generalization. In the revision, we will make this explicit in the problem setup by directly tying the phrase “alignment bottleneck” to _total capacity_ and briefly explaining that it captures the combined effect of cognitive limits and interface bandwidth in the channel $U \to H \to Y$.

---

> ### Author Response · Authors · 2025-11-26
> **Response to Reviewer 2NMe (3/3)**
>
> **Weakness 5: Is the analysis presented in the paper is for parametrized settings or it does not matter, a discussion would help?**
>
> Our results are formulated at an abstract, model-agnostic level. The Fano-style lower bound is a minimax statement over all decoders and does not assume any parameterization. The PAC–Bayes part is phrased in terms of the mutual information between the data and the learned hypothesis $\theta$; $\theta$ can index a parametric neural network, a non-parametric method, or even a randomized algorithm. In the revision, we will add a short clarification that the bounds apply to **any** alignment procedure that learns only from the feedback $Y$, regardless of parameterization, and that analyzing specific optimizers (PPO, DPO, GRPO, etc.) is an interesting but separate follow-up direction.
>
> **Weakness 6: Would the analysis of this work could shed some light or guidance on the type of feedback one should use for alignment, such as preference feedback? Is it optimal?**
>
> Our analysis works at the level of the abstract channel $U \to H \to Y$, where different feedback formats correspond to different ways of mapping $H$ into $Y$, and therefore to different effective capacities $\overline{C}_{\text{tot}\mid S}$. We do not claim that any particular format (such as binary pairwise preferences) is optimal. Instead, the bounds show that when the value complexity $\log M$ of the rubric is high, **low-bandwidth channels are structurally limiting**: even with a good rubric and more data, they cannot transmit enough information about $U$ to reliably distinguish configurations. The main design guidance implied by our work is therefore: for complex constitutions, improving alignment requires either (i) simplifying or modularizing the rubric, or (ii) increasing the effective capacity of the feedback interface (for example, via richer scales, structured critiques, or decomposed evaluations). We will make this guidance explicit in the implications/discussion section.
>
> **Weakness 7: There are no empirical results in the paper. I understand that the work in theoretical in nature, but is it possible to provide some basic experiments to may be just connect what is the lower bound, and how we are doing currently with the available methods to motivate the importance of these lower bounds.**
>
> We agree that it is important to connect the lower bounds to practice. In response, we have added a controlled synthetic DPO experiment, summarized in the global comment **Empirical Observation of Capacity Limits**. We use Qwen2.5-3B as the policy and Qwen2.5-7B as both oracle $U$ and rater $H$ under a five-rule constitution, and we compare a high-capacity channel (prompt enforces all rules) with a bottlenecked channel (prompt collapses evaluation to cheap agreeableness/tone heuristics). As we increase the dataset size $m$, the oracle scores under the high-capacity channel improve roughly monotonically, while under the bottlenecked channel they saturate quickly and eventually fall below the base model, with the model becoming more sycophantic. **We have included a concise version of this table and curve in the revised paper (**Section 8**)** and explicitly connected them to the “capacity wall” predicted by our theory.

---

> > ### Comment · Reviewer_2NMe · 2025-11-27
> >
> > I would like to thank the authors for the responses in rebuttal. Based on the discussion provided, I would like to maintain my positive evaluation of the work and keep my current rating. Thank you.

---

> > > ### Author Response · Authors · 2025-12-03
> > > **Thank you for your continued support**
> > >
> > > We sincerely thank Reviewer 2NMe for the constructive discussion and for maintaining the positive evaluation. We appreciate you pointing us to the relevant literature on bounded rationality, which helped contextualize our contribution. We have now uploaded the revised PDF, which incorporates the new experimental results  and the expanded discussion you suggested.

---

### Official Review · Reviewer_Yk7L · 2025-11-02

**Soundness:** 2
**Presentation:** 2
**Contribution:** 2
**Rating:** 2
**Confidence:** 4

**Summary:**

The paper argues that the primary alignment problems in large language models come from a fundamental information bottleneck between humans and models treating human feedback as passing through a limited channel with a fixed capacity.  The paper builds this argument using simple information-theoretic tools (like Fano’s and PAC–Bayes bounds) to show both lower and upper performance limits that depend on this same “human capacity.”

**Strengths:**

- The paper provides an interesting way to connect cognitive science and information theory to alignment in a clean, mathematical way.
- Gives a natural explanation for problems like reward hacking which happens when models overfit beyond what the human feedback can represent.
- Highlights that just collecting more data alone won’t fix alignment unless human feedback capacity increases.

**Weaknesses:**

- The analysis and the conception is interesting however is not directly measurable or testable in real LLM pipelines.
- The analysis builds on standard tools of information theory primarily under the constant “capacity” for humans assumption. However, the connection is weak and is not clear what are the key new aspects coming out from the analysis? This makes it hard to evaluate the paper
- Can the authors provide empirical demonstration - even a toy example of the hypothesis? Also, explaining what are the key-terms that are new or novel to the LLM/Agent Alignment paradigm? Even connecting with relevant papers and showing which term in the bound is new or provide some insights is crucial.

Note : I am open to update my view after understanding the key new aspects of this understanding.

**Questions:**

See above.

---

> ### Author Response · Authors · 2025-11-26
> **Response to Reviewer Yk7L (1/2)**
>
> We thank the reviewer for the careful and critical assessment and for pressing on measurability, novelty and empirical support, which are central for this kind of work.
>
> **Weakness 1: The analysis and the conception is interesting however is not directly measurable or testable in real LLM pipelines.**
>
> In the paper and in the global comment **Operationalizing $U$ as Constitutions and Rubrics**, we instantiate $U$ as a finite constitution or rubric codebook, so that value complexity $\log M$ and the effective feedback capacity $\overline{C}_{\text{tot}\mid S}$ are tied to concrete objects. In practice, one can probe the channel $U \to H \to Y$ by introducing a strong oracle that applies the full rubric and comparing its judgments to the labels produced by time limited human raters or model raters through the actual feedback interface. On a held out calibration set, the agreement pattern between oracle labels and production labels then provides an estimate or lower bound on the effective capacity of the feedback process. The synthetic Qwen 2.5 experiment described in **Empirical Observation of Capacity Limits** illustrates this measurement idea in a controlled setting: we vary the rater prompt to create a high capacity channel and a bottlenecked channel and observe that increasing dataset size helps in the former but saturates and then hurts in the latter, matching the predicted capacity wall behavior.
>
> **Weakness 2: The analysis builds on standard tools of information theory primarily under the constant “capacity” for humans assumption. However, the connection is weak and is not clear what are the key new aspects coming out from the analysis? This makes it hard to evaluate the paper.**
>
> Our intention is to make explicit a structural constraint on feedback alignment that is usually only discussed informally. The key conceptual step is to model the full data generating pipeline for feedback alignment as a single noisy channel, from the underlying rubric configuration $U$ through a bounded rational rater state $H$ to the logged feedback $Y$, with $U$ realized as a finite codebook of rubric outcomes. On top of this channel model we show that one and the same quantity, the effective feedback capacity $\overline{C}_{\text{tot}\mid S}$, both sets a floor on achievable alignment performance through a Fano style argument and controls the usable complexity in a PAC Bayes generalization bound. This produces an alignment performance interval in which the lower and upper sides are coupled by the same capacity term, rather than treating label quality and model complexity as independent knobs. In the revision we will make this coupling and its implications more explicit in alignment language, as described in the global comment **Primary Contribution: The Capacity Coupled Structure**.

---

> ### Author Response · Authors · 2025-11-26
> **Response to Reviewer Yk7L (2/2)**
>
> **Weakness 3: Can the authors provide empirical demonstration, even a toy example of the hypothesis? Also, explaining what are the key terms that are new or novel to the LLM/Agent Alignment paradigm? Even connecting with relevant papers and showing which term in the bound is new or provide some insights is crucial.**
>
> To address the empirical part, we have added a synthetic preference training experiment with Qwen 2.5 models, detailed in **Empirical Observation of Capacity Limits** and included in **Section 8** of the revised PDF. A five rule constitution defines $U$. Qwen2.5 7B Instruct acts as both the oracle that scores responses under this constitution and as the rater under two prompts, one high capacity prompt that tries to enforce all rules and one bottlenecked prompt that mainly follows tone and agreeableness heuristics. Qwen2.5 3B Instruct is trained by DPO on increasing amounts of preference data from each channel. Under the high capacity channel, the oracle score improves roughly monotonically with more data. Under the bottlenecked channel, the oracle score shows the predicted saturation followed by degradation, eventually dropping below the base model as the policy becomes more sycophantic. This directly instantiates channel overfitting and the capacity wall in a concrete LLM setting.
>
> Regarding what is new for alignment, our main contribution is a fully specified and, in principle, computable theoretical framework that models the entire feedback alignment pipeline as an information channel from the rubric level objective $U$ through the bounded rater state $H$ to the labels $Y$. This moves beyond analysing optimization on a fixed dataset of labels and instead captures how limitations arise from the total information that can be transmitted from the constitution to the data. Within this framework we highlight two concrete quantities. The value complexity $\log M$ makes the richness of a fixed constitution or rubric explicit. The total feedback capacity summarizes how much of that rubric can reach the learner through bounded raters and a given feedback interface. These quantities jointly define the regimes in which feedback alignment can succeed or fail, and appears on both sides of the alignment performance interval, linking lower bounds on alignment error to upper bounds on usable complexity. **In the revision we have connected this framework and these terms more explicitly to existing feedback alignment and bounded rationality work and made clear which parts are standard information theory and which are specific to this setting.**

---

### Official Review · Reviewer_Uxb5 · 2025-11-03

**Soundness:** 3
**Presentation:** 2
**Contribution:** 3
**Rating:** 4
**Confidence:** 1

**Summary:**

In this paper, the author presents a theoretical framework explaining why feedback-based alignment methods for LLMs lead to failures like sycophancy and reward hacking, despite scaling. The central idea is that the human feedback loop is a resource-limited information channel, imposing a fundamental alignment bottleneck. The paper models this process as a two-stage cascade: $U \rightarrow H \rightarrow Y \text{ given } S$ (Value $\rightarrow$ Judgment $\rightarrow$ Feedback). This channel is limited by a finite cognitive capacity ($\overline{C}_{\text{tot}|S}$) due to bounded rationality. The authors argue that this single-capacity coupling leads to key implications: 1. Scaling dataset size alone is insufficient to overcome the bottleneck. 2. Aligning on more complex values requires a corresponding increase in channel capacity. 3. Once this capacity is saturated, a powerful optimizer will fit the rater's biases, providing a theoretical explanation for reward hacking and sycophancy.

**Strengths:**

1. Motivated by concepts from bounded rationality, the author provides a principled framework that reframes alignment failure as an information-channel limit.

2. In the Alignment Performance Interval bound, the coupling of the Fano lower bound and the PAC-Bayes upper bound using the single channel capacity term ($\overline{C}_{\text{tot}|S}$) is interesting.

**Weaknesses:**

1. The paper is motivated entirely by theoretical analysis and does not present experiments to validate the theoretical claims. A synthetic experiment to support the claims would make the paper stronger.

2. The framework primarily relies on the average total capacity, $\overline{C}_{\text{tot}|S}$. However, the paper provides no discussion on how one might estimate or measure this quantity in a real-world setting.

**Questions:**

Please refer weaknesses

---

> ### Author Response · Authors · 2025-11-26
> **Response to Reviewer Uxb5**
>
> We thank the reviewer for the careful assessment and for highlighting concrete ways to strengthen the paper.
>
> **Weakness 1: The paper is motivated entirely by theoretical analysis and does not present experiments to validate the theoretical claims. A synthetic experiment to support the claims would make the paper stronger.**
>
> We appreciate this suggestion. We have added a controlled DPO alignment experiment with Qwen 2.5 models, described in the global comment **Empirical Observation of Capacity Limits**. We use Qwen2.5 3B Instruct as the policy, and Qwen2.5 7B Instruct as both the oracle $U$ and the rater $H$ under a five rule constitution. We instantiate a High Capacity channel that tries to enforce all rules versus a Bottlenecked channel that relies on compressed tone and agreeableness heuristics. In the High Capacity case, performance under the oracle score improves roughly monotonically as we increase the dataset size. In the Bottlenecked case, we see the predicted saturation and degradation: oracle performance is essentially flat at small data sizes ($50$--$500$ pairs), and finally drops to 39.59 at $1000$ pairs (below the base model score of 40.88) as the policy overfits the limited heuristics and becomes more sycophantic. This matches the capacity wall behavior predicted by our framework. **A concise version of this experiment has been included in the revised paper (Section 8)**, and we will release the code and synthetic data.
>
> **Weakness2: The framework primarily relies on the average total capacity, $\overline{C}_{\text{tot}\mid S}$. However, the paper provides no discussion on how one might estimate or measure this quantity in a real-world setting.**
>
> We agree the original draft did not say clearly enough how to connect this quantity to practice. In the Fano style formulation used in the paper, $U$ is instantiated as a finite constitution or rubric codebook that records which rules are satisfied for each example, and $H$ is the rater’s bounded judgment under time and attention limits. In concrete feedback alignment pipelines, one can probe this channel by introducing a strong oracle, for example a constitution tuned model or expert labels that apply the full rubric, and comparing its judgments to the labels that are actually logged.
>
> Operationally, the idea is simple. First, fix a specific constitution or rubric and a gold standard oracle that follows it as closely as possible. Second, collect oracle decisions and production labels on the same held out examples, where the production labels come from human raters or model raters using the actual feedback interface such as pairwise preferences or numeric scores. Third, use this calibration set to measure how much of the oracle’s judgment can be recovered from the logged labels, for example through consistency statistics or mutual information style estimates. This does not give an exact number for $\overline{C}_{\text{tot}\mid S}$, but it provides a practical way to estimate or lower bound the effective capacity of a given feedback process. We will add a short, concrete discussion of this estimation picture in the revision and highlight it in the theory sections referenced in the global comment **Operationalizing $U$ as Constitutions and Rubrics**.

---

### Author Response · Authors · 2025-11-21
**Primary Contribution: The Capacity Coupled Structure**

We thank the reviewers for their thoughtful questions about this work. Our main contribution is **an information-theoretic model of the _entire_ feedback-alignment pipeline**: we treat alignment as a capacity-limited channel $U \to H \to Y \mid S$, where $U$ is a finite constitution/rubric codebook, $H$ is the rater’s bounded-rational internal judgment, and $Y$ is the logged feedback, making both the target values and the cognitive bottleneck explicit and, in principle, estimable via the value complexity $\log M$ and total feedback capacity $\overline C_{\text{tot}\mid S}$. On top of this model, using standard Fano-style and PAC–Bayes tools, we obtain a **capacity-coupled alignment interval** in which the same feedback capacity $\overline C_{\text{tot}\mid S}$ simultaneously controls a Fano lower bound on true risk and a PAC–Bayes upper bound on generalization, tying label noise and hypothesis complexity to a single physical bottleneck: the number of bits about values that can traverse the feedback channel.

This interval yields a **capacity-determined lower bound on alignment risk**: when the value complexity $\log M$ is large relative to $\overline C_{\text{tot}\mid S}$, the risk is bounded away from zero and **cannot** be driven to zero by increasing the dataset size $m$, formally explaining the saturation effects observed in alignment and contrasting with standard scaling views that assume monotonic gains from more data. The same capacity term also provides a mechanism for sycophancy and reward hacking: once the informative signal about $U$ saturates $\overline C_{\text{tot}\mid S}$, further optimization inevitably fits residual channel regularities rather than the intended value, **making sycophancy a structural consequence of capacity-limited feedback rather than an algorithmic bug**.

Taken together, the model and interval deliver a clear design message: for complex constitutions (large $\log M$), low-bandwidth feedback is structurally insufficient regardless of data volume, so progress requires either reducing value complexity or increasing effective feedback capacity through richer interfaces, decomposed evaluations, or stronger raters.

---

### Author Response · Authors · 2025-11-21
**Operationalizing U as Constitutions and Rubrics**

Our lower-bound and structural analysis **instantiates $U$ in a discrete, codebook-like way** that provides a rigorous theoretical model for alignment.

Practical alignment pipelines rarely operate on “all human values” in the abstract. Instead, they use **constitutions, safety policies, and detailed rubrics**: finite lists of rules and criteria. Our formalization **offers a natural instantiation of this by treating the value space as a finite set of $M$ distinguishable configurations.** Each configuration corresponds to a pattern of rule satisfaction for a given context–response pair (for example, which of the constitution rules are satisfied or violated). The random variable $U$ is simply an index into this finite set.

From an information-theoretic perspective, this finite set is exactly a codebook, and the “value complexity” is $\log M$, the number of bits needed to identify the configuration. The important point for readers is not the terminology, but the consequence: **once we fix a constitution or rubric, $U$ becomes a concrete, discrete, and operational object.** Increasing the richness of the rubric, for example by adding more rules or finer distinctions, increases $M$ and therefore increases the value complexity that the feedback channel must support.

Within this picture:

- $U$ is the underlying rubric / constitution configuration that is correct for the example.

- $H$ is the rater’s effective rubric and attention pattern under bounded time and cognitive resources. A high-capacity rater $H_{\text{full}}$ tries to attend to all rules (truthfulness, harmlessness, instruction-following, clarity, anti-sycophancy). A bottlenecked rater $H_{\text{limited}}$ collapses to cheap heuristics such as “does this sound agreeable and fluent?”.

- $Y$ is the observable preference label or score that is actually logged and used for RLHF or DPO.


Because $U$ is modeled as a finite codebook, capacity is not purely abstract. In principle, one can approximate or lower-bound $\overline{C}_{\text{tot}\mid S}$ by measuring mutual information between a gold-standard proxy for $U$ (for example, expert labels or a strong oracle’s judgment according to the full rubric) and the rater outputs $Y$ on held-out data. This is a natural direction for future work: using our framework as a yardstick for estimating how much of a given rubric actually makes it through a real feedback process.

---

### Author Response · Authors · 2025-11-21
**Bounded Rationality as the Basis for Channel Limits**

We argue that **bounded rationality is the physical mechanism that necessitates a channel capacity model.**

In standard alignment theory, the label $Y$ is often treated as a "ground truth" corrupted only by random noise. Our work challenges this by asking: **Why is the feedback signal imperfect?** The answer lies in bounded rationality: humans possess finite computational resources.

This cognitive constraint forces the human rater to perform **lossy compression** on the task:

- **The Cognitive Bottleneck $U \to H$:** A rater cannot evaluate a response against a complex, multi-dimensional constitution $U$ in real-time. Instead, they must "satisfice" by adopting a simplified mental model $H$, attending only to the most salient features (e.g., tone or length) while ignoring subtle errors.

- **The Information Result:** This simplified mental model mathematically corresponds to a finite mutual information $I(U; H) < H(U)$.


Therefore, **"Capacity" in our paper is the quantifiable measure of the rater's bounded rationality**. If humans were unbounded rational agents, the channel capacity would be infinite, and our "Capacity Wall" would disappear. The information-theoretic bounds are thus the direct mathematical consequence of the cognitive assumptions.

**Relation to Recent Work:** We thank the reviewer for highlighting the recent work by Chehade et al. titled Bounded Rationality for LLMs: Satisficing Alignment at Inference-Time (arXiv:2505.23729). This work applies satisficing principles to the inference stage of language models. Specifically, they develop a decoding strategy that optimizes a primary objective while meeting threshold constraints on secondary ones. In contrast, our work focuses on the human feedback mechanism during the training stage. We model the cognitive limits of human raters as an information bottleneck within the feedback channel itself. While Chehade et al. operationalize bounded rationality to design an inference algorithm, we utilize this concept to derive fundamental information-theoretic bounds on alignment learnability. These two lines of research address distinct stages of the alignment pipeline and are complementary in nature.

---

### Author Response · Authors · 2025-11-21
**Empirical Observation of Capacity Limits**

To address the request for empirical validation, we have conducted and integrated a controlled synthetic experiment testing the prediction that data scaling can harm alignment under capacity limits. Using a Qwen2.5-3B policy and Qwen2.5-7B as both Oracle ($U$) and Rater ($H$), we define capacity via the rater's prompt. This setup operationalizes RLAIF bottlenecks where models struggle to attend to complex constitutions irrespective of data volume.

We compared a **High-Capacity Channel ($H_{\text{high}}$)**, where the rater prompt explicitly enforces a five-rule constitution, against a **Bottlenecked Channel ($H_{\text{low}}$)**, where the prompt compresses the objective into heuristics for emotional validation and politeness. We trained DPO policies on dataset sizes $m \in \{50, 200, 500, 1000\}$ generated by each channel and evaluated them against the Oracle ($u_{\text{total}} \in [0, 50]$).

**Table 1: Mean Oracle Constitution Score ($u_{\text{total}}$)**

|**Dataset Size (m)**|**High-Capacity (Hhigh​)**|**Bottlenecked (Hlow​)**|
|---|---|---|
|**Base (m=0)**|**40.88**|**40.88**|
|**m = 50**|41.68|41.69|
|**m = 200**|41.75|41.53|
|**m = 500**|42.57|41.64|
|**m = 1000**|**43.08**|**39.59**|

The results confirm the theoretical "capacity wall." Under the high-capacity channel, performance improves monotonically with data size, reaching a mean oracle score of 43.08 at $m=1000$. In contrast, the bottlenecked channel shows immediate saturation followed by degradation. Scores remain flat around 41.6 until $m=500$, but at $m=1000$, performance drops sharply to 39.59, falling well below the base model. Qualitative inspection reveals that this drop corresponds to channel overfitting: the model learns the rater's specific bias toward sycophancy at the expense of the underlying truthfulness rules. These findings are now included as **Section 8** in the revised manuscript.

---

### Author Response · Authors · 2025-11-21
**Revision Plan**

We commit to the following revisions:

- **Integrate the “Capacity Wall” experiment** described above into the main text, explicitly connecting the two theoretical regimes (capacity sufficient vs. capacity insufficient) to the observed behaviors: stable improvement under $H_{\text{full}}$ and inverted-$U$ degradation under $H_{\text{limited}}$.

- **Strengthen the operational mapping** inside the existing theory sections. In Section 3 (Problem Setup) and Section 4 (Information-Theoretic Lower Bounds), we will explicitly interpret $U$ as a discrete constitution / rubric codebook with complexity $\log M$, $H$ as the rater’s effective rubric and attention pattern under bounded rationality, and $Y$ as the logged feedback label. Around the Fano codebook construction, we will add brief inline explanations connecting the abstract codebook to practical rubric-based evaluations and sketch how one could estimate or lower-bound $\overline{C}_{\text{tot}\mid S}$ from oracle–rater mutual information.

- **Clarify the role of bounded rationality in the setup and related work**, emphasizing that it is the structural assumption that induces the information constraint $U \to H \to Y$, and we will clearly distinguish our training-time feedback bottleneck from inference-time compute limits in other work.

- **Expand the implications discussion to give concrete design guidance**: when binary preferences (about one bit per sample) are structurally inadequate for a given constitution, when richer feedback interfaces (critiques, decomposed tasks, richer scales) are needed to increase effective capacity, and how rough estimates of $\overline{C}_{\text{tot}\mid S}$ could inform choices of KL budgets and optimization strength to avoid channel overfitting.

- **Make the exposition more accessible by simplifying notation where possible and adding brief intuitive explanations and examples around key results**, so that the final version is easier to follow and more directly connects the information-theoretic bounds with the accompanying empirical illustration of capacity-limited alignment.

---

### Author Response · Authors · 2025-12-03
**Major Revision Uploaded: New Experiments and Full Text Update**

We want to thank the Area Chair and all reviewers for the time spent assessing our work. We focused on addressing your primary concerns regarding empirical validation and the connection to engineering practice. A substantially revised PDF has been uploaded.

The updated manuscript now includes a new experimental section using Qwen-2.5 to verify our theoretical predictions. We also updated the paper throughout to explicitly map our variables to real-world alignment concepts like constitutions and rater bandwidth. All major edits are highlighted in blue text for easy identification. We genuinely appreciate the constructive criticism as it pushed us to bridge the gap between theory and practice and significantly strengthened the paper.

Best regards,

The Authors

---

### Meta-Review · Area_Chair_ujbB · 2026-01-09

**Summary:**

The paper studies how limited human feedback would affect the theoretical limit of LLM alignment. The main contribution is to model the human feedback loop as a two-stage channel and establish a two-sided error bound the depends on the average capacity of the channel.

The main concerns from the reviewers are the relevance of the proposed framework to the real-world alignment regime, the practical implication of the proposed framework, and lack of empirical demonstration. While the paper includes experiments on how limitation on the feedback channel would affect the alignment performance, the message from the results is mainly a replication of know phenomena on the impact of imperfect feedbacks. It is still unclear how the proposed quantity would bring new insights on practical alignment regimes.

Hence, I would recommend a rejection for the paper. The draft would be significantly stronger if the authors could discuss how the provided channel capacity quantity could be evaluated in practice and how it can be used to guide alignment practice.

**Reviewer Concerns:**

The authors addressed most concerns from the reviewers on the clarity of the paper. However, as discussed in the meta-review, the authors didn't provide a satisfying justification on how the work provides new insights on alignment practice.

**Reviewer Scores:**

I would expect most reviewers to keep their score.

---

### Decision · Program_Chairs · 2026-01-26

Reject